# Genetic variation in IL-4 activated tissue resident macrophages determines strain-specific synergistic responses to LPS epigenetically

Mingming Zhao [1], Dragana Jankovic [1], Verena M. Link [2], Camila Oliveira Silva Souza [1], Katherine M. Hornick [3], Oyebola Oyesola [1], Yasmine Belkaid [2], Justin Lack [3] & Png Loke [1] ✉

How macrophages in the tissue environment integrate multiple stimuli depends on the genetic background of the host, but this is still poorly understood. We investigate IL-4 activation of male C57BL/6 and BALB/c strain specific in vivo tissue-resident macrophages (TRMs) from the peritoneal cavity. C57BL/6 TRMs are more transcriptionally responsive to IL-4 stimulation, with induced genes associated with more super enhancers, induced enhancers, and topologically associating domains (TAD) boundaries. IL-4-directed epigenomic remodeling reveals C57BL/6 specific enrichment of NF-κB, IRF, and STAT motifs. Additionally, IL-4-activated C57BL/6 TRMs demonstrate an augmented synergistic response upon in vitro lipopolysaccharide (LPS) exposure, despite naïve BALB/c TRMs displaying a more robust transcriptional response to LPS. Single-cell RNA sequencing (scRNA-seq) analysis of mixed bone marrow chimeras indicates that transcriptional differences and synergy are cell intrinsic within the same tissue environment. Hence, genetic variation alters IL-4-induced cell intrinsic epigenetic reprogramming resulting in strain specific synergistic responses to LPS exposure.

Tissue-resident macrophages (TRMs) play crucial roles in maintaining tissue homeostasis and responding to inflammation[1–3]. Like many TRMs, large peritoneal cavity TRMs are predominantly derived from embryonic progenitors seeded into tissues before birth and maintained by local proliferation[4]. Dysfunction of TRMs can cause severe and fatal developmental disorders, as well as dysregulated inflammatory responses[5,6]. The effects of genetic variation on signal-dependent gene expression in bone marrow derived macrophages (BMDMs)[7] has revealed inbred mouse strain-specific responses to stimulus, but strain-specific differences in TRMs remains poorly characterized.

Understanding the diversity of TRMs response to activation stimuli across different genetic backgrounds may offer new insights into determinants of immune variation.

During type 2 immune response, the cytokines IL-4 and IL-13 signal through IL-4Rα to trigger a macrophage activation state (M(IL-4) or alternatively activated M2 macrophages)[8] that promotes control of helminth infection[9] and tissue repair[10,11]. Importantly, different tissue macrophages will respond differently to IL-4 activation in vivo and we previously found that tissue-resident macrophages and inflammatory macrophages derived from monocytes are phenotypically different

[1]Type 2 Immunity Section, Laboratory of Parasitic Diseases, National Institute of Allergy and Infectious Diseases (NIAID), National Institutes of Health (NIH), Bethesda, MD, USA. [2]Metaorganism Immunity Section, Laboratory of Host Immunity and Microbiome, National Institute of Allergy and Infectious Diseases, National Institutes of Health, Bethesda, MD, USA. [3]NIAID Collaborative Bioinformatics Resource, Integrated Data Sciences Section, Research Technology Branch, Division of Intramural Research, NIAID, NIH, Bethesda, MD, USA. ✉e-mail: Png.Loke@nih.gov

following IL-4 activation[3,12]. IL-4 drives the synthesis and activation of signal-dependent transcription factors (SDTFs)[13], which are mainly IRF4 and STAT6, while the SDTFs induced after LPS stimuli include IRF1, IRF5, IRF8, STAT1, STAT2 and NF-κB[14]. However, the exact SDTF repertoire may vary depending on differences in the initial cellular state prior to stimulation. The epigenomic and transcriptional programs of TRM responses to inflammatory signals are tightly and dynamically regulated by a combination of transcription factors including lineage-determining transcription factors (LDTFs) and SDTFs. Sequence variants can disrupt the collaborative binding of LDTFs such as PU.1 with SDTFs such as STAT6[15]. Motif mutation analysis of IL-4 activated BMDMs highlights the critical role of LDTFs, including PU.1, C/EBPβ, and AP-1 family members, as well as SDTFs such as STAT6 and PPARγ, in IL-4-driven transcription[16]. Importantly, the IL-4 activation state of macrophages is not a fixed differentiation state and can be converted to a classical activation state in vitro by subsequent treatment with IFNγ/LPS[17]. Recently, IL-4 primed BMDMs have been shown to undergo "extended synergy" with LPS treatment[18]. However, how genetic variation influences the epigenomic changes resulting from IL-4 activation and subsequent exposure to toll-like receptor (TLR) ligands such as LPS remains unexplored. Furthermore, such systematic examinations of transcriptional regulation have thus far focused on in vitro derived BMDMs, which may be distinct from in vivo derived and activated TRMs.

Here, we uncovered how IL-4 activation can remodel the epigenomic organization differently in TRMs from BL/6 and BALB/c mice. IL-4 activation in BL/6 TRMs results in enhanced accessibility for NF-κB and IRF activity, as defined by ATAC-seq and H3K27ac ChIP-seq (promoter-proximal) or Hi-C (promoter-distal) analysis. This BL/6 specific NF-κB enrichment resulted in an enhanced synergistic effect with subsequent LPS exposure on IL-4-activated BL/6 TRMs compared with BALB/c TRMs. Hence, the background genetic variation can affect the integration of different stimuli within TRMs due to different epigenomic reorganization.

## Results

To better understand how genome structure and enhancer-promoter interactions affect transcriptional responses to IL-4 activation in vivo in tissue resident macrophages, we integrated multi-modal genomic data on F4/80^Hi large peritoneal macrophages from BL/6 and BALB/c mice after treatment with IL-4-Fc intraperitoneally (i.p.). Analyses included RNA-seq, ATAC-seq, ChIP-seq for H3K27ac, genome-wide chromatin conformation capture (Hi-C), scRNA-seq, and scATAC-seq (Fig. 1a). All data generated are available in the Gene Expression Omnibus (GEO) series GSE248038.

### Transcriptome analysis reveals strain-specific differences in IL-4-induced gene expression

To evaluate strain specific transcriptional responses to IL-4 stimulation in vivo, RNA-seq was performed on F4/80+ purified peritoneal TRMs from individual mice (n = 3/group) and principal components analysis (PCA) showed sample replicates clustering tightly together (Fig. 1b). Strain specific differences on the PC1 axis explains more variance (61%) than the effects of IL-4 stimulation on the PC2 axis (34%). Notably, the difference between control (Ctrl) and IL-4 treated TRMs along the PC2 axis is greater for BL/6 than the BALB/c strain (Fig. 1b). The top 10 genes that are driving PC1 for BL/6 TRMs (Fig. 1c, left) include *Gvin1*, *Marco* and *Fabp5*. The top 10 genes that are driving PC2 (Fig. 1c, right) include genes associated with alternative activation (e.g., *Chil3, Retnla,* and *Arg1*), but also include other genes (e.g., *Rnase2a, Col1a2, and Col3a1*). We next identified genes upregulated by IL-4 treatment in each strain (FDR ≤ 0.05; ≥2-fold) relative to TRMs from naive untreated mice (Ctrl) and found a total of 530 genes upregulated in BALB/c and 1,227 genes upregulated in BL/6. Of those genes 304 overlapped between strains (Fig. 1d), resulting in 226 BALB/c and 923 BL/6 genes

uniquely responding to IL-4 in the different strains. This result indicates that BL/6 peritoneal TRMs are more transcriptionally responsive to IL-4 activation than BALB/c TRMs. Since BL/6 and BALB/c TRMs responses could be shaped by differences in the tissue environment, we also examined transcriptional profiles from BMDMs activated in vitro by IL-4[16] using the same bioinformatics analysis parameters (Fig. 1e). We identified 171 overlapping genes upregulated by IL-4 treatment in both strains. Additionally, BL/6 macrophages exhibited an additional 280 strain-specific upregulated genes, while BALB/c macrophages had 108 strain-specific upregulated genes (Fig. 1e). These results indicate that at least some component of BL/6 macrophages being more responsive to IL-4 stimulation than BALB/c should be cell intrinsic. However, BL/6 and BALB/c TRM transcripts can be expressed at different levels under baseline conditions, hence defining only upregulated genes in each strain may miss aspects of strain specific IL-4 mediated transcriptional regulation. We used a multifactor DESeq2 model (further details in the method) to identify strain specific differentially regulated genes (FDR ≤ 0.05) after IL-4 activation and *k-means* clustering to divide 3 main patterns of transcriptional changes (C1, C2, and C3), of which we focused on IL-4 upregulated genes (C2 + C3) (Fig. 1f). 2006 BL/6 specific genes are significantly upregulated, of which 513 have more than 2-fold higher expression compared to naive macrophages (Ctrl). For BALB/c, there are 1205 strain specific upregulated genes, of which only 237 genes have more than 2-fold higher expression compared to naive macrophages (Ctrl). Figure 1g displays a heatmap illustrating the fold-change values compared with Ctrl for genes list with more than a 2-fold change from Fig. 1f (C2 + C3). Additionally, Supplementary Fig. 1a (left) and Supplementary Fig. 1a (right) show heatmaps of normalized reads for these genes specific to BL/6 and BALB/c, respectively. Quantified read counts indicate that BL/6-specific genes are expressed at significantly higher levels compared to BALB/c-specific genes (Supplementary Fig. 1b). Examples of IL-4 induced genes that are not strain specific include *Chil3* (Fig. 1h), whereas *Trem2*, which is important in Alzheimer's disease and cancer[19,20], is an IL-4 induced gene specific for BL/6 macrophages (Fig. 1i). These results illustrate the divergent IL-4 transcriptional responses between BL/6 and BALB/c macrophages, with BL/6 macrophages being more transcriptionally responsive to IL-4 activation than BALB/c macrophages.

Next, we performed motif enrichment analysis on the promoter regions (−400bp to +100 bp of transcription start site (TSS)) of BL/6 and BALB/c strain specific genes (Fig. 1g). De novo motif analysis identified the RELB motif (Fig. 1j) in BL/6-specific IL-4 activated genes which were induced more than 2-fold compared to Ctrl (Fig. 1j, left). No motifs were enriched in promoters of BALB/c-specific IL-4 activated genes (Fig. 1j, right). RELB constitutes one subunit of NF-κB family[21]. This implies that the genes upregulated specifically in BL/6 in response to IL-4 are more regulated by NF-κB signaling.

Helminth infections induce IL-4 production and *Heligomosomoides polygyrus (H. polygyrus)* infection can also induce the accumulation of IL-4 activated TRMs in the peritoneal cavity[22]. Hence, we compared the natural accumulation of peritoneal TRMs from *H. polygyrus* infected BL/6 and BALB/c mice. By RNA-seq, we found that the expanded BL/6 TRMs from *H. polygyrus* infected mice also have a stronger transcriptional response compared to infected BALB/c mice. 469 genes were uniquely upregulated (FDR ≤ 0.05; ≥2-fold) in BL/6 *H. polygyrus* TRMs relative to TRMs from untreated mice, compared to 269 in infected BALB/c mice (Fig. 1k). By *k-means* clustering on the differentially regulated genes after DESeq2 analysis, we found 963 (FDR ≤ 0.05) strain specific genes upregulated (C2 + C3) in BL/6 *H. polygyrus* induced TRMs and only 545 (FDR ≤ 0.05) strain specific genes upregulated (C2 + C3) in BALB/c *H. polygyrus* induced TRMs (Fig. 1l). We analyzed the promotor regions of the whole genes list that are strains specific upregulated compared to naïve TRMs (FDR ≤ 0.05, ≥2-fold) for motif enrichment. IRF motifs are enriched in genes

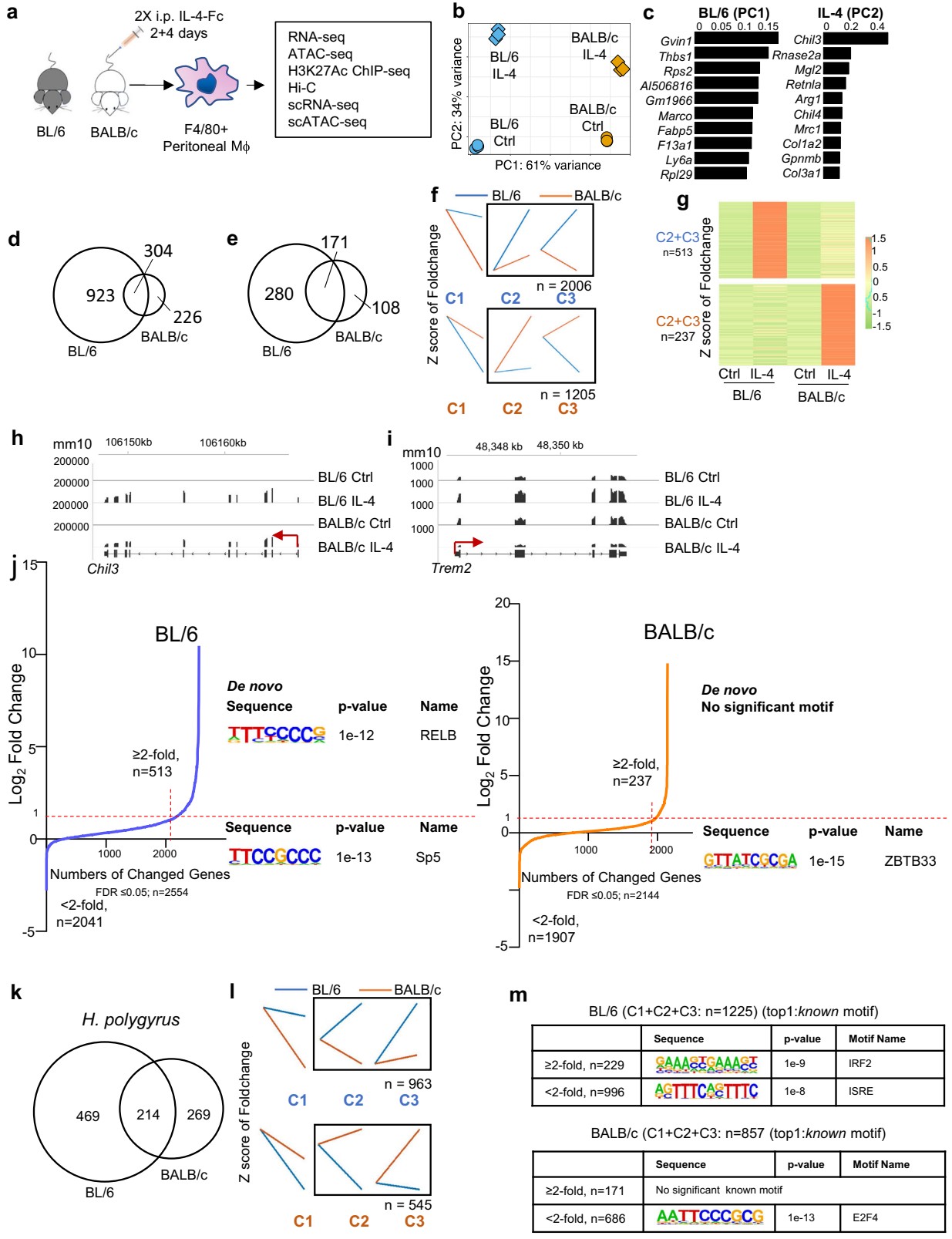

specifically upregulated in BL/6 *H. polygyrus* infected TRMs, but there were no significant motif enrichments in genes specially upregulated in BALB/c *H. polygyrus* TRMs (Fig. 1m). Lastly, we compared the transcriptional responses to IL-4-Fc and *H. polygyrus* and observed some overlap in the significantly induced genes; however, *H. polygyrus* infection induces a greater number of unique genes, in both BL/6 and BALB/c strains (Supplementary Fig. 1c). Notably, *H. polygyrus* induces a

more distinct transcriptional phenotype in macrophages compared to naïve peritoneal macrophages, likely due to the complex stimuli from the entire parasite response, in contrast to the singular influence of IL-4-Fc. This complexity results in the enrichment of IRF motifs in the promoters of infection-induced genes in BL/6 macrophages, though it does not preclude the induction of genes containing the NF-κB motif. Indeed, our analysis of the 180 genes with NF-κB motifs that are

**Fig. 1 | Response to IL-4 is more pronounced in peritoneal TRMs from BL/6 mice compared to BALB/c mice. a** Overview of the experimental design and the primary main datasets for the IL-4 treatment. **b** Principal Component Analysis (PCA) plot illustrates the relationship between replicates of RNA-seq samples taken from TRMs after in vivo IL-4 treatment in both BL/6 and BALB/c mice. Principal Component 1 (PC1) captures a substantial 61% of the data's variation, while Principal Component 2 (PC2) contributes to 34%. **c** The first list (left) is based on the first principal component (PC1) and highlights genes that are specifically enriched in BL/6 mice. The x axis value represents the -loading value of PC1. The second list (right) is derived from the second principal component (PC2) and represents genes that are enriched after IL-4 treatment. The x axis value represents the loading value of PC2. **d** Overlap of genes significantly induced (FDR ≤ 0.05, ≥2-fold) identified by EBSeq method after IL-4 treatment in peritoneal TRMs from BL/6 and BALB/c mice. **e** Overlap of genes significantly induced (FDR ≤ 0.05, ≥2-fold) identified by EBSeq method after IL-4 treatment in BMDMs from BL/6 and BALB/c mice. **f** Differentially expressed genes were identified by DESeq2 (FDR ≤ 0.05). Each group was further separated into three kinetic patterns by k-means clustering. The names of each cluster and the numbers of genes in black square are shown. Clusters labeled shown in blue are more induced in BL/6 mice, while those in orange are more induced in BALB/c mice. **g** Heatmap displaying fold-change compared to Ctrl, showing the BL/6 and BALB/c-specific induced genes (FDR ≤ 0.05 and ≥2-foldchange compared with Ctrl) in the C2 + C3 group in (**f**). Each row represents the z-score of the fold-change. **h, i** Representative IGV browser tracks from RNA-seq analyses of commonly induced between strains (Chil3) and BL/6-specific induced (Trem2) genes in Ctrl and IL-4-treated cells. mm10 annotations are shown. **j** Transcription factor motifs found at the promoter region (-400bp to +100 bp from TSS) of selected genes in BL/6 (Left) or BALB/c (Right) specific upregulated genes compared to the Ctrl, determined by HOMER (findMotifs.pl). De novo motifs identified by HOMER are labeled on the right side of the figure. **k** Overlap of genes significantly induced (FDR ≤ 0.05, ≥2-fold) identified by EBSeq method after H. polygyrus infected in TRMs from BL/6 and BALB/c mice. **l** Kinetic patterns of differentially expressed genes in H. polygyrus-infected BL/6 and BALB/c mice: Differentially expressed genes were identified using DESeq2 (FDR ≤ 0.05). Each group was further divided into three kinetic patterns via k-means clustering, and the number of genes in selected clusters is displayed. **m** Motif analysis at the promoter region (−400 bp to +100 bp from TSS) of BL/6 or BALB/c specific regulated genes was conducted using HOMER, and only known motifs are shown.

---

induced by IL-4-Fc in BL/6 macrophages reveals that these genes are also induced by *H. polygyrus*, but not in BALB/c mice (Supplementary Fig. 1d). Additionally, when we compared IL-4 upregulated genes in TRMs and BMDMs, there is considerable variation in responses between both strains and macrophage subtypes indicating differences in IL-4 responsiveness by compartment as well as genetic background (Supplementary Fig. 1e).

In summary, these results indicate that peritoneal TRMs from BL/6 mice exhibit a heightened type 2 transcriptional response compared to BALB/c, regardless of whether driven by IL-4 treatment or *H. polygyrus* infection. This heightened response is likely cell intrinsic, as this observation was confirmed in BMDMs activated in vitro. Additionally, motif analysis of the promoter region of induced genes indicated that NF-κB and IRF transcription factors may play a more substantial role as SDTFs in BL/6 TRMs following IL-4 activation in vivo than in BALB/c TRMs.

## IL-4 activation results in strain-specific epigenetic states with distinct transcription factor accessibility in the enhancer landscape

To investigate the effect of IL-4 activation on the chromatin structure in TRMs, we next performed ATAC-seq to measure open chromatin and enhancers, and H3K27ac ChIP-seq to measure enhancer activation (Fig. 2). First, we identified constitutively accessible regions which are not altered in accessibility in response to IL-4 treatment of which 20,188 regions were shared between strains and 18,336 were uniquely accessible in BALB/c mice (3731 in BL/6 mice, respectively) (Fig. 2a). We further identified accessible regions that are inducible by IL-4 activation of which only 1407 regions were shared between strains and 4801 regions were uniquely accessible in BL/6 (3660 in BALB/c mice, respectively) (Fig. 2b). Notably, BALB/c macrophages have more strain specific constitutive regions (n = 18,336 in BALB/c versus n = 3731 in BL/6), whereas there is only a small difference in IL-4 inducible accessible regions (n = 4801 in BL/6 versus n = 3660 in BALB/c) (Fig. 2a, b). Additionally, induced accessible regions from BL/6 macrophages are enriched in intron and intergenic regions, whereas constitutive accessible regions are enriched in promoters (Fig. 2c). However, this contrast between induced and accessible regions is less pronounced in BALB/c macrophages (Fig. 2c). Hence, BL/6 macrophages are more responsive to IL-4 driven chromatin remodeling than BALB/c macrophages, whereas BALB/c macrophages have more constitutively accessible regions.

To identify transcription factors associated with IL-4 driven chromatin remodeling, we performed transcription factor motif enrichment analysis with HOMER on the IL-4 induced ATAC-seq peak regions (Fig. 2d). This revealed significant enrichment of LDTFs such as

Sfpi1 (PU.1) and AP1(Cebpd, Fols2), as expected in both strains. Notably, IRF and NF-κB SDTF motifs are significantly enriched in BL/6 specific IL-4 induced peaks, but not in BALB/c specific induced regions. In contrast, the homeobox motif (HOXA13) is only enriched in BALB/c induced ATAC peaks (Fig. 2d). These results suggest that chromatin remodeling after IL-4 activation in BL/6 TRMs is driven by IRF and the NF-κB family of transcription factors (TFs), while in the BALB/c background the bZIP and homeobox family of TFs are the driving factors. We next used chromVAR[23] to identify enriched motifs that are strain specific and IL-4 responsive and grouped by hierarchical clustering (Supplementary Fig. 2a, left). This analysis suggests that the putative activity level of PU.1, NF-κB, IRFs and STATs are enriched in IL-4 activated BL/6 TRMs, whereas AP-1(bZIP motif) activity is enriched in BALB/c TRMs under both baseline as well as IL-4 activated conditions (Fig. 2e). Other bZIP motif clusters, including C/EBP motifs such as C/EBPA, C/EBPB and C/EBPD were also more enriched in BALB/c TRMs after IL-4 treatment (Supplementary Fig. 2a, right). These findings indicate that IL-4-induced chromatin remodeling enhances transcription factor accessibility differently in BL/6 and BALB/c TRMs and for BL/6 TRMs these include LDTFs like PU.1, along with SDTFs such as NF-κB, IRFs, and STATs.

Next, we applied H3K27ac ChIP-seq to study change of activated enhancers following IL-4 stimulation. We observed 141 enhancers uniquely activated in BL/6 and 347 enhancers uniquely activated in BALB/c (Supplementary Fig. 2b), which showed different functional enrichment by using the Genomic Regions Enrichment of Annotations Tool (GREAT) on BL/6 and BALB/c specific enhancers (Supplementary Fig. 2c). De novo motif analysis on strain-specific enhancer regions revealed that IRFs were exclusively enriched in BL/6-specific enhancers, whereas the homeobox motif (*Meis*) showed enrichment in BALB/c (Supplementary Fig. 2d), which is consistent with other analysis (Fig. 2d). To identify strain-specific IL-4-induced enhancers, we combined H3K27ac ChIP-seq and ATAC-seq analysis. Initially, we overlapped the IL-4-induced ATAC peaks with H3K27ac ChIP-seq data in both strains, allowing us to identify strain-specific induced enhancers (Fig. 2f). We identified 2,718 enhancers induced in IL-4 stimulated BL/6 TRMs and 1,457 enhancers induced in IL-4 stimulated BALB/c TRMs. Notably, the majority of these IL-4 induced enhancers are strain specific, with only 561 shared between strains (Fig. 2g). PU.1 and IRF motifs (PU.1:IRF8) were more enriched in BL/6, whereas bZIP motifs (cEBP-like) were more enriched in BALB/c (Fig. 2g). We associated enhancers with genes specifically upregulated in BL/6 and BALB/c TRMs (Fig. 1g) by applying the enhancers to the nearest promoters of genes using the HOMER and found that the enhancers near the BL/6 specific upregulated genes (promoter-proximal genomic regions)

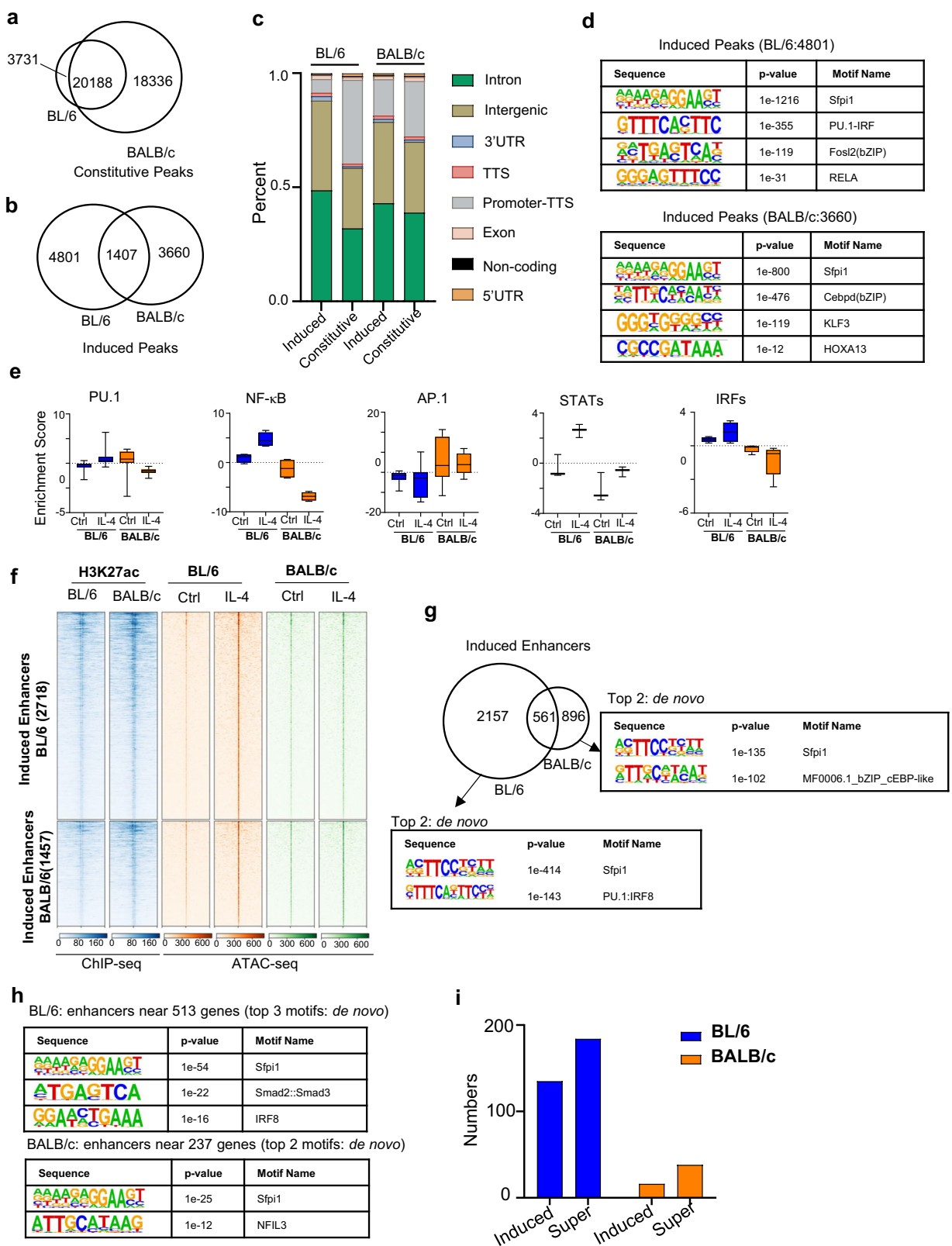

are more enriched for SDTFs motifs (e.g., IRFs) (Fig. 2h). When we mapped BL/6 and BALB/c specific IL-4 upregulated genes to the nearby strain specific induced enhancers (Fig. 2i), we found that out of 513 BL/6 specific genes, 135 are associated with induced enhancers, whereas only 16 out of 237 BALB/c specific genes are associated with induced enhancers. Additionally, when we defined super-enhancer regions by H3K27ac ChIP-seq using HOMER[24], after annotating the

super-enhancer with the nearest promoter of gene, we found that 184 out of 2,006 BL/6 specific IL-4 upregulated genes are associated with super enhancers nearby whereas only 38 BALB/c specific genes are associated with super enhancers. However, the number of total super-enhancer regions in BL/6 ($n = 578$) and BALB/c ($n = 555$) macrophages are quite similar. These results may partially explain the increased IL-4 transcriptional responsiveness of BL/6 TRMs, as IL-4 upregulated

**Fig. 2 | Differential chromatin-level response to IL-4 in BL/6 and BALB/c mice.** **a** Constitutive ATAC-seq peaks (FDR > 0.001) are identified by DiffBind when comparing control (Ctrl) and IL-4 treatment conditions in both strains. The Venn diagram visually represents the overlap of these constitutive ATAC-seq peaks between BL/6 and BALB/c mice. **b** ATAC-seq induced peaks (FDR ≤ 0.001) are identified by DiffBind between Ctrl and IL-4 treatment in BL/6 and BALB/c mice. Venn diagram shows the overlap of the induced ATAC-seq peaks from BL/6 and BALB/c mice. **c** Genomic locations of ATAC-seq peaks, both constitutive and induced by IL-4 treatment, display binding sites across multiple genomic regions as annotated by HOMER. The strain names are labeled above the bars. **d** Using HOMER, we identified the top 4 transcription factors enriched in induced peaks in BL/6 (left panels) and BALB/c (right panels). Only de novo motifs are showed here. Significance was calculated using default statistical setting provided by HOMER. **e** Box and whiskers plot shows transcription factor motif enrichment scores in ATAC-seq data for select motifs that exhibit differential enrichment between BL/6 and BALB/c, assessed using ChromVAR. PU.1 ($n = 17$ relevant motifs); NF-κB ($n = 4$ relevant motifs); AP.1 ($n = 30$ relevant motifs); STATs ($n = 3$ relevant motifs); IRFs ($n = 5$ relevant motifs). Each motif value represents the average of motif enrichment scores of the two replicates. Data are presented as mean value of motif enrichment scores from all motifs +/− SEM. **f** A heatmap displaying induced enhancers derived from induced ATAC-seq peak data. **g** Overlap of significantly induced enhancers after IL-4 treatment in peritoneal TRMs from two strains. The top two de novo motifs identified by HOMER are enriched at strain-specific induced enhancers, with an observed gain in the IRF motifs in BL/6 mice. Significance was calculated using default statistical setting provided by HOMER. **h** The top 3 (BL/6) and top 2 (BALB/c) de novo motifs identified by HOMER enriched in enhancers near strain-specific induced genes (FDR ≤ 0.05 and ≥ 2-foldchange compared with Ctrl). Significance was calculated using default statistical setting provided by HOMER. **i** The count of induced enhancers or super enhancers near strain-specific induced genes.

genes in BL/6 TRMs are more associated with induced enhancers and super enhancer regions.

## IL-4 primes BL/6 TRMs to synergize with lipopolysaccharide (LPS) signaling more effectively than BALB/c TRMs

Recently, IL-4 priming was shown to increase NF-κB-p65 binding resulting in "extended synergy" of IL-4-polarized BMDMs upon lipopolysaccharide (LPS) exposure[18]. Since we observe that in vivo IL-4 activation in BL/6 TRMs enhances accessibility to motifs associated with SDTFs such as NF-κB and IRFs, we hypothesized that synergy between IL-4 priming and LPS exposure would also be more pronounced in BL/6 TRMs relative to BALB/c TRMs. Hence, we compared the in vitro (3 h) LPS response of TRMs isolated from Ctrl and IL-4 in vivo treated BL/6 and BALB/c mice by RNA-seq (Supplementary Fig. 3a). PCA shows that the transcriptional response to IL-4 + LPS exposure is more distinct from LPS treatment in BL/6 TRMs compared to BALB/c TRMs (Fig. 3a). However, the distance on the PC1 axis between LPS exposed and untreated TRMs is greater on the BALB/c background (Fig. 3a), indicating that the transcriptional effects of LPS on BALB/c TRMs may be greater. We next identified differentially expressed genes (DEGs) (FDR ≤ 0.05 and ≥2-fold) between untreated vs LPS treated TRMs (LPS vs Ctrl), as well as between LPS treated TRMs and LPS treated IL-4 activated TRMs (IL-4 + LPS vs LPS) on both backgrounds (Fig. 3b). The majority of DEGs ($n = 863$) between Ctrl versus LPS exposure in BL/6 TRMs are also affected in BALB/c TRMs. Consistent with the PCA plot, more genes are differently upregulated between Ctrl and LPS in BALB/c macrophages ($n = 831$) than BL/6 ($n = 178$) indicating a stronger transcriptional response to LPS exposure compared to naive peritoneal BALB/c TRMs (Fig. 3b, left). However, in line with our observation in the PCA plot (Fig. 3a), when we compared the response of LPS treated IL-4 activated TRMs with LPS treated naïve TRMs (Fig. 3b, right), we found that more genes are upregulated in BL/6 ($n = 909$) compared to BALB/c TRMs ($n = 113$) with an only small number of genes affected in both strains ($n = 161$). GO analysis revealed that 'metabolite process' was associated with the BALB/c-specific upregulated genes (LPS vs Ctrl, $n = 831$), while 'immune system process' was linked to the BL/6-specific upregulated genes (IL-4 + LPS vs LPS, $n = 909$) (Supplementary Fig. 3b).

Next, we employed *k-means* clustering to group the differentially regulated genes between IL-4 + LPS and LPS alone, as identified by DESeq2 using fold-change values across all samples relative to LPS treatment (Fig. 3c). Additionally, an alternative *k-means* analysis based on fold-change values relative to Ctrl is presented in Supplementary Fig. 3c. Both analyses revealed the presence of six distinct clusters. Notably, cluster 2 (C2:1361) and cluster 4 (C4:1030) exhibited a marked increase in LPS inducibility in IL-4-primed BL/6 TRMs compared to LPS treatment only (Fig. 3c), and cluster 2 exhibited a synergistic response following IL-4 + LPS treatment (Fig. 3d). From the 1361 genes in Cluster 2 (C2), 568 genes showed a greater than 2-fold change in BL/6, while in

Cluster 4 (C4), 266 genes met the filtering criteria. In contrast, among the 754 genes in Cluster 6 (C6), only 37 exhibiting a 2-fold difference between IL-4 + LPS compared to LPS treatment only in BALB/c macrophages. This demonstrated that extended synergy of IL-4-polarized TRMs upon LPS exposure is much more pronounced in the BL/6 genetic background. Six representative genes of cluster C2 are shown in Fig. 3e. Of these genes, *Edn1* and *Ccl2* have previously been reported upregulated in IL-4-primed and LPS-exposed BL/6 BMDMs[18]. Induction of *Nfkbia* by LPS (Supplementary Fig. 3d, left) and *Chil3* by IL-4 (Supplementary Fig. 3d, right) in both BL/6 and BALB/c macrophages showing that signaling pathways downstream of IL-4 and LPS are functionally active in both strains. GO analysis on the C2 and C4 gene clusters showed that the majority of immune response genes were concentrated in the C2 cluster, whereas C4 genes were predominantly associated with metabolic processes (Supplementary Fig. 3e). Other proinflammatory genes (*Il1a, Il1b, Il6, and Tnf*) show no synergistic effect in either mouse strain, indicating only a subset of genes exhibit a synergistic response like those in the C2 cluster (Supplementary Fig. 3f).

To elucidate the epigenomic state of IL-4-primed TRMs at the regulatory elements governing genes in C2 and C4, we examined the ATAC-seq signal at these loci. For both C2 and C4 genes, the promoter-proximal genomic regions, as defined by adjacent ATAC-seq peaks, exhibited enhanced ATAC-seq signals in IL-4-primed BL/6 TRMs. Notably, this heightened accessibility was only observed in BL/6 IL-4-treated TRMs, in contrast with BALB/c TRMs where such induced accessibility was not evident. This highlights a greater accessibility in BL/6 IL-4-treated TRMs compared to BALB/c TRMs in the promoter-proximal genomic regions of the synergistically regulated genes (C2) (Fig. 3f). The accessible regulatory elements at the promotor-proximal regions of C2 and C4 genes are enriched for the Sfpi1 (PU.1) and IRF motifs (Fig. 3g), and C2 genes have a significant enrichment of the NF-κB motif. The whole top 20 motif list are showed in Supplementary Fig. 3g. These results indicate that IL-4 treatment in BL/6 modified the epigenome to increase accessibility for the IRF and NF-κB family of TFs, which may mediate the extended synergy of IL-4 primed LPS transcriptional responses. However, this epigenomic conditioning for synergistic responses does not occur in the BALB/c background.

## Hi-C analysis identifies NF-κB binding sites in the promoter-distal regions associated with BL/6 specific IL-4 induced genes

Regulatory elements can govern gene expression across extensive genomic distances, exerting influence on genes situated far from promoters—termed here as promoter-distal regions[25]. To define promoter-distal regions and investigate 3D genome structure, we performed Hi-C on IL-4 activated BL/6 and BALB/c peritoneal TRMs. We examined the role of long-range interactions of regulatory elements in controlling strain-specific IL-4 transcriptional responses

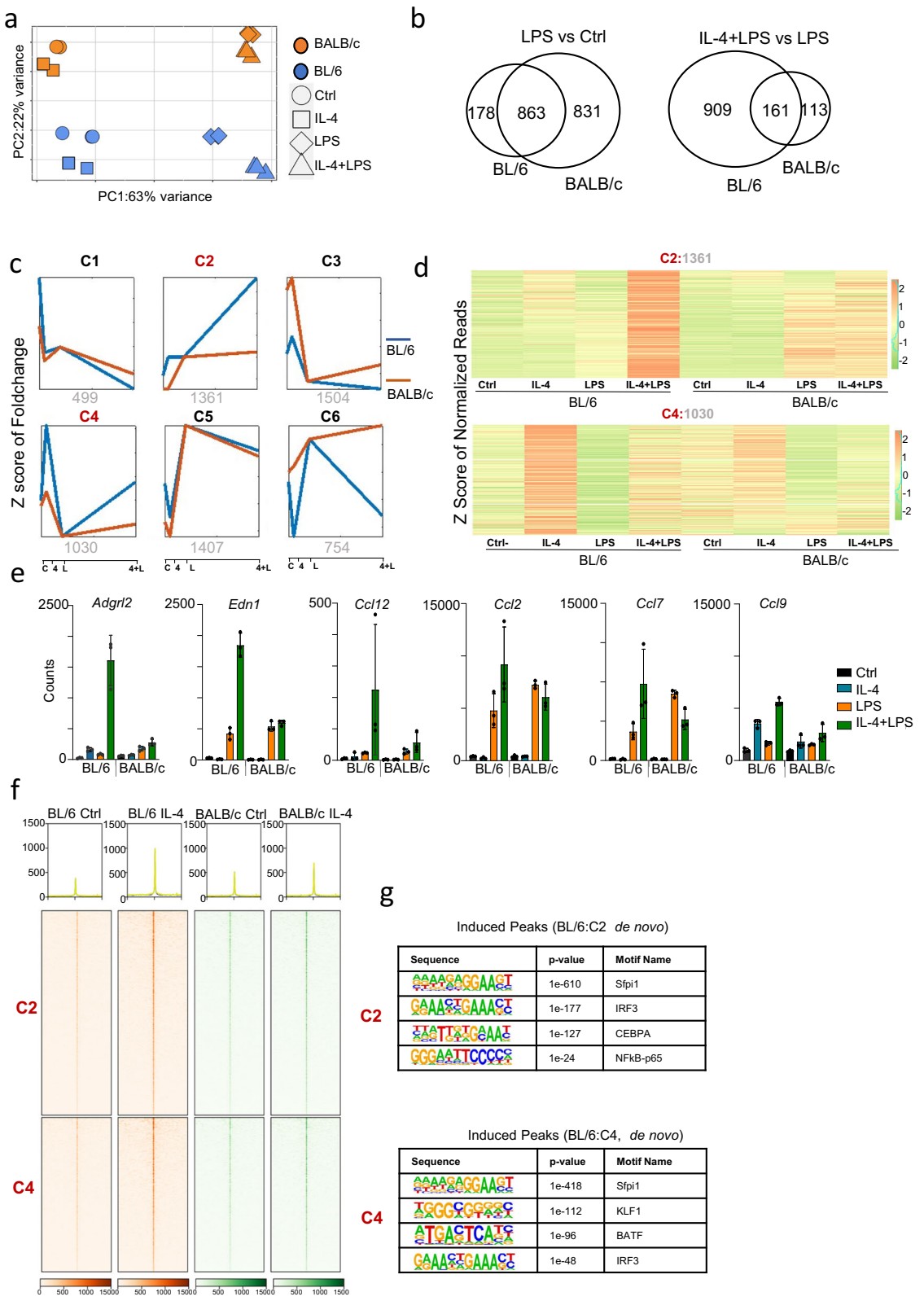

through 3D interactions. We first combined Hi-C analysis with ATAC-seq data to define the promoter based regulatory elements. We determined the promotor-distal regulatory elements for BL/6 specific and BALB/c specific IL-4 upregulated genes, and next identified TF motifs enriched in these accessible regulatory elements. We found 535 promoter-distal regions were related to the BL/6 specific upregulated genes (513) and 143 promoter-distal regions were related to BALB/c specific upregulated genes (237). Motif analysis by HOMER revealed enrichment of IRF motifs (de novo) and NF-κB motifs (*known*) in promoter-distal regions specific to BL/6 (Fig. 4a, left), but not in BALB/c promoter-distal regions (Fig. 4a, right). These findings indicate that NF-κB and IRFs may play a role in the regulation of IL-4 upregulated genes specific to BL/6 through long-range promoter-distal interactions.

**Fig. 3 | Synergistic activation was observed after LPS treatment in IL-4-primed TRMs from both strains, with a more pronounced synergistic response in BL/6 mice. a** Principal component analysis (PCA) of RNA-seq data from IL-4, LPS, IL-4 + LPS, and Ctrl mice (*n* = 3 per group) shows the primary separation of samples between IL-4 + LPS and LPS treatment in BL/6 mice. **b** Significantly induced genes were identified by EBSeq in two comparisons (LPS vs Ctrl and LPS + IL-4 vs LPS). The left panel shows the overlap of genes significantly induced (FDR ≥ 0.05, ≥2-fold) after LPS treatment compared to Ctrl in macrophages from both strains. The right panel shows the overlap of genes significantly induced (FDR ≤ 0.05, ≥2-fold) after IL-4 + LPS treatment compared to LPS treatment in macrophages from both strains. **c** Differentially expressed genes were identified by DESeq2 (FDR ≤ 0.05) that changed in response to any treatment between strains. The differentially expressed genes were then further separated into 6 kinetic patterns *by k-means* clustering

based on the fold-change compared to LPS treatment. The names of clusters (top:C1-C6) and the number of genes (bottom) in each cluster are shown. Activation is noted on the x-axis, where "C" indicates Ctrl, "4" indicates IL-4 treatment, "L" indicates LPS treatment, and "4 + L" indicates IL-4 and LPS cotreatment. **d** The heatmap represents normalized gene expression levels in clusters C2 and C4 from (**c**) in both strains. The number in each group is labeled at the top of the heatmap. Each row represents the z score of normalized reads. **e** Bar graph showed the RNA-seq counts on the selected genes after IL-4 or LPS treatment (*n* = 3 biological replicates). Data are presented as mean values +/− SEM. **f** Heatmap produced by deepTools showed read distribution plot of ATAC-seq occupancy near genes in cluster C2 and C4 from (**c**) in both strains. **g** Motif analysis by HOMER showed IRF and NF-κB motifs under the ATAC-seq peaks near the genes in C2 and C4 from (**c**). Significance was calculated using default statistical setting provided by HOMER.

Utilizing MEME motif analysis[26], we found that 18 out of 237 BALB/c specific IL-4 upregulated genes have NF-κB motifs in the distal regulatory regions (-7%), whereas 81 out of 513 BL/6 specific IL-4 upregulated genes have NF-κB motifs on the proximal-distal regulatory regions (-15%). For example, *Trem2* and *Wdfy3* are only upregulated in BL/6 TRMs after IL-4 activation (Figs. 1i, 4b). In the accessible promoter-distal regions for these genes, we only found NF-κB and IRF motifs in BL/6 TRMs but not in BALB/c TRMs, which may partially explain why these two genes are only upregulated in BL/6 TRMs. We compared the expression levels of genes (*n* = 180) containing an NF-κB motif in either the promoter or promoter-distal regions among the 513 genes specifically upregulated by IL-4 exclusively in BL/6 TRMs (Fig. 1g) with genes lacking NF-κB motifs (*n* = 332). Notably, the expression level for genes with NF-κB motifs is higher than those that are not associated with NF-κB motifs (Fig. 4c). Also, the level of induction by IL-4 compared to Ctrl is higher for genes associated with NF-κB motifs in BL/6 TRMs (Fig. 4c). Hence, NF-κB may play a role in BL/6 TRMs specific promoter-distal transcriptional regulation during IL-4 activation in vivo.

Subsequently, we investigated the interplay between 3D genome organization and genes that undergo upregulation upon IL-4 priming followed by LPS exposure in BL/6 TRMs (Fig. 3c, C2 and C4 gene clusters), in comparison to BALB/c TRMs. We identified the promoter-distal regions of C2 and C4 genes through DNA interactions obtained from Hi-C analysis, and MEME analysis was employed to pinpoint promoter-distal regions containing the NF-κB motif. Our findings revealed that 310 (out of 1361) genes from C2 (-23%) and 209 genes (out of 1030) from C4 (-20%) are regulated by promoter-distal regions associated with the NF-κB motif. Motif enrichment analysis conducted with HOMER in these regions demonstrated a significant enrichment of the NF-κB motif, verifying the region selection made by MEME (Supplementary Fig. 4a). The ATAC-seq data revealed that the accessible regulatory elements within the promoter-distal regions of genes belonging to both C2 and C4 from Fig. 3c are also the regions that experience increased accessibility following IL-4 activation in BL/6 TRMs (Supplementary Fig. 4b). For instance, the promotor distal region of *Hbegf*, which is important in would healing[27] from cluster C2, has a NF-κB motif under BL/6 specific loop that may partly explain the synergistic LPS response (Supplementary Fig. 3c, d). Hence, the heightened transcriptional activity upon LPS activation appears to be associated with IL-4 priming-induced chromatin remodeling in BL/6 TRMs. These findings imply the presence of a preexisting and comparatively stable landscape of BL/6-specific enhancer-promoter interactions in macrophages, whose regulatory function becomes activated in response to IL-4.

To investigate the durability of epigenetic changes, we isolated BL/6 peritoneal macrophages 16 days after IL-4-Fc treatment and performed ATAC-seq to determine whether the changes in chromatin accessibility were long-lasting. In contrast to the 4801 induced differential peaks observed 4 days after IL-4-Fc treatment compared to naïve control macrophages, we found no statistically significant

differential peaks (FDR ≥ 0.05) at 16 days post-treatment and the epigenetic profile is indistinguishable from control macrophages by PCA (Supplementary Fig. 4e, left). Additionally, regulatory elements containing the NF-κB motif after IL-4-Fc treatment were no longer accessible following 16 days of IL-4 treatment in specific examples of synergistic genes, such as Ccl2 and Ccl7 (Supplementary Fig. 4e, right). This indicates that the BL/6 synergistic response to LPS stimulation is not long lasting.

## Genes activated by IL-4 in BL/6 TRMs exhibit closer proximity to topologically associating domain (TAD) boundaries and display a more homogeneous expression pattern

Hi-C data also provides information on TADs. TADs are genomic regions that form units of three-dimensional (3D) nuclear organization whereby DNA physically interacts with each other, which can regulate gene expression through enhancer-promoter interactions within TADs[28]. We used the Juicer pipeline and TAD caller arrowhead[29] to annotate TADs in BL/6 (*n* = 2362) and BALB/c (*n* = 2227) macrophages (Fig. 5a–b). In line with previous investigations, the majority (*n* = 1941) of TAD regions are shared between BL/6 and BALB/c macrophages. However, notable differences exist, such as at the *Chil3* locus, where gene expression is higher in BL/6, both at the basal level and after IL-4 treatment (Fig. 5b, Supplementary Fig. 3d, right).

TAD boundaries (see methods) delineate the regions that lie between TADs[30]. Upon mapping IL-4-induced genes to TAD boundary regions, we observed that the promoters of BL/6-specific upregulated genes (*n* = 2006, Fig. 1f) are positioned closer to TAD boundaries compared to BALB/c-specific genes (*n* = 1205) (Fig. 5c, left). Conversely, non-IL-4-induced genes exhibiting strain-specific differences only at basal levels do not show significant distinctions in the distance to TAD boundaries between BL/6 and BALB/c naïve macrophages (Fig. 5c, right). Earlier studies have demonstrated that the mean expression of genes within TAD boundaries is higher than that of genes located outside these boundaries. Additionally, genes within boundary regions exhibit lower variability in expression compared to those situated in non-boundary regions[31]. Here, we also observed that BL/6-specific IL-4 upregulated genes (*n* = 2006) exhibit higher expression levels, as indicated by normalized read counts, compared to BALB/c-specific IL-4 upregulated genes (*n* = 1205) (Fig. 5d). Furthermore, to analyze transcriptional variability between strains, we calculated the Fano factor (a measure of gene expression noise[32]), and observed a lower Fano factor for BL/6-specific genes upregulated after IL-4 activation compared to BALB/c-specific upregulated genes (Fig. 5e). This aligns with the notion that gene expression variability tends to be lower in genes situated near TAD boundary regions. Using Weighted Gene Co-expression Network Analysis (WGCNA)[33] to identify modules of co-expressed genes, we found that BALB/c macrophages exhibited a higher number of modules (Fig. 5f), contrasting with BL/6 macrophages where a singular large module prevailed. Hence, there is more variability in the expression of genes in the BALB/c macrophages.

**a**

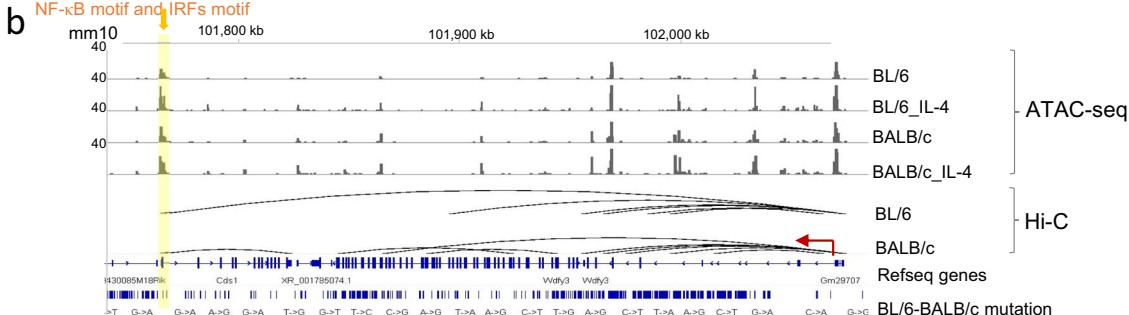

BL/6:promotor-distal for 513 genes (top 4 motif: *de novo*)

| Sequence | p-value | Motif Name |
|---|---|---|
| | 1e-62 | Sfpi1 |
| | 1e-48 | BORIS |
| | 1e-16 | CEBPD |
| | 1e-15 | PU.1-IRF |

BALB/c: promotor-distal for 237 genes (top 2 motif: *de novo*)

| Sequence | p-value | Motif Name |
|---|---|---|
| | 1e-26 | BORIS |
| | 1e-14 | PU.1 |

BL/6:promotor-distal for 513 genes (NF-κB motif: *known*)

| Sequence | p-value | Motif Name |
|---|---|---|
| | 1e-4 | NFkB-p65 |

No NF-κB motif enriched

**b**

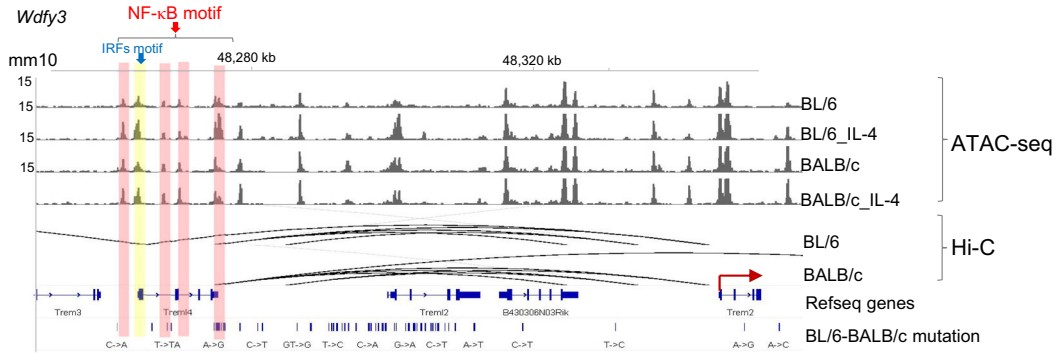

*Wdfy3*

*Trem2*

**c**

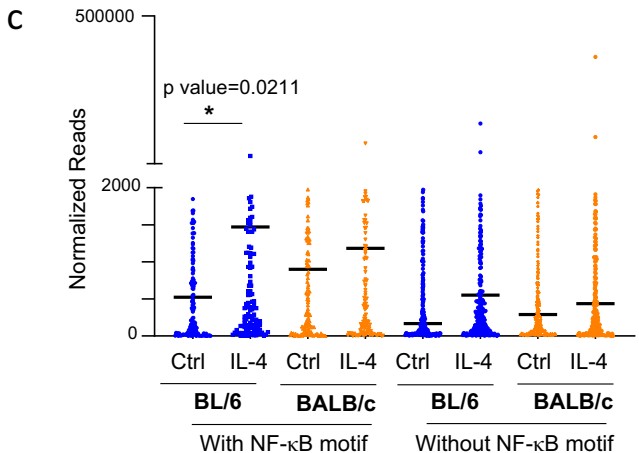

When we analyzed strain-specific IL-4 induced genes (Fig. 1g, BL/6: *n* = 513 and BALB/c: *n* = 237), we observed that the promoter regions of BL/6-specific IL-4 upregulated genes (*n* = 513) were closer to TAD boundaries compared to those in BALB/c mice (Supplementary Fig. 5a), and, after IL-4 activation, the BL/6-specific genes exhibited higher expression values and lower variability, as indicated by the Fano factor (Supplementary Fig. 5b, c).

In summary, our findings suggest that IL-4-induced genes specific to BL/6 TRMs tend to be located closer to TAD boundaries, exhibiting higher expression levels and lower variability. Conversely, BALB/c-specific IL-4-activated genes are situated further away from TAD boundaries with lower expression levels and higher variability. These differences may contribute to the stronger transcriptional response to IL-4 activation in BL/6 TRMs.

**Fig. 4 | Promoter-distal regions of IL-4-induced genes in BL/6 TRMs revealed NF-κB motifs through the analysis of chromatin loop features obtained from Hi-C analyses. a** Top transcription factor binding motifs enriched within promoter-distal regions defined by chromatin loops from Hi-C in strain-specific induced genes. De novo motifs were identified by HOMER in promoter-distal regions from BL/6-induced genes (left panels) and BALB/c-induced genes (right panels). Notably, the *known* motif enriched in promoter-distal regions of BL/6-induced genes includes NF-κB. Significance was calculated using default statistical setting provided by HOMER. **b** Genome browser tracks (IGV) illustrate examples of promoter-distal regulatory element enriched with IRF and NF-κB motifs linked to the promoter regions of BL/6-specific induced genes. The solid yellow bars represent ATAC-seq peaks around the chromatin loop region from Hi-C containing both IRF

and NF-κB motifs, while the solid pink bars represent ATAC-seq peaks around the chromatin loop region from Hi-C with only the NF-κB motif. **c** We divided BL/6-specific induced genes (FDR ≤ 0.05 and ≥ 2-fold) into two groups. One group includes genes with the NF-κB motif in either the promoter site (−400bp to +100 bp from TSS) or promoter-distal regions defined by chromatin loops from Hi-C (*n* = 180), while the other group has no NF-κB motif in either the promoter site or promoter-distal regions defined by chromatin loops from Hi-C (*n* = 333). The scatter plot shows that genes with the NF-κB motif in promoter or promoter-distal regions exhibit higher expression and a greater induction compared to Ctrl in BL/6 mice. The y-axis represents normalized RNA-seq reads. *p*-values were calculated using an unpaired t-test, two-sided.

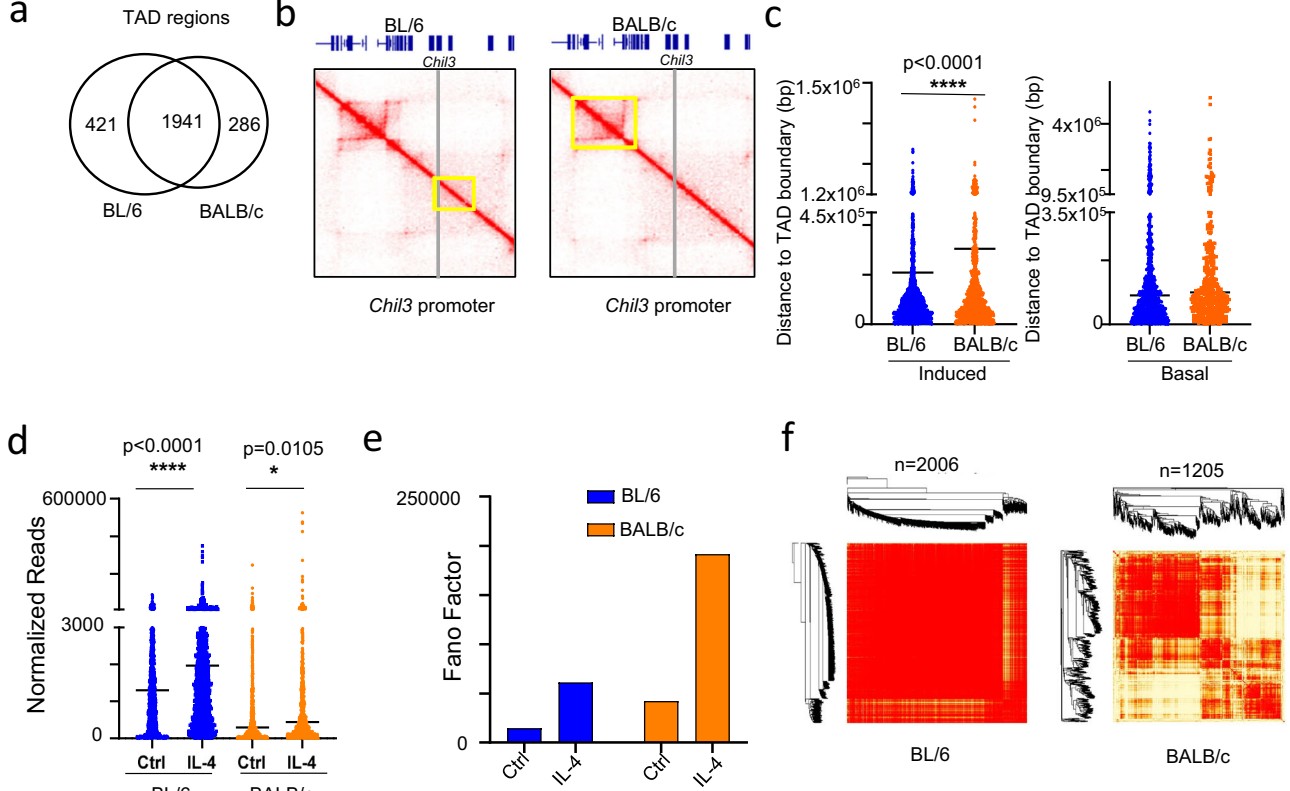

**Fig. 5 | Genes induced by IL-4 in BL/6 TRMs exhibit a stronger association with the boundaries of TADs identified through Hi-C analysis. This association is characterized by higher gene expression values and lower expression variability. a** Venn diagram illustrates the genome-wide co-localization TADs identified by Hi-C analysis in both strains. **b** Hi-C maps from TRMs of BL/6 and BALB/c mice exhibit a representative TAD region that differs between strains (25 kb resolution; raw count map). The grey line indicates the promoter region of the *Chil3* gene, and the yellow square indicates the TAD region from Hi-C data. **c** Dot plots display the distance of the promoter regions of strain-specific induced genes to the TAD boundary region (left). The strain-specific induced genes are identified by DESeq2 in Fig. 1f. Only genes meeting the threshold of FDR ≤ 0.05 (2006 in BL/6 and 1205 in BALB/c mice) are used. Differential genes between BL/6 and BALB/c at the basal level were identified using the EBSeq method with the threshold of FDR ≤ 0.05

(right). The TAD boundaries are defined in the methods and materials section. Asterisks (****) indicate *p*-values < 0.0001. *p*-values were calculated using an unpaired t-test, two-sided. Data are presented as mean ± standard deviation. **d** Scatter plots displaying the RNA expression levels of strain-specific induced genes (2006 in BL/6 and 1205 in BALB/c mice) with corresponding *p*-values placed at the top of the figure. Asterisks (****) indicate p-values < 0.0001; Asterisks (*) indicate p-values = 0.0105. *p*-values were calculated using a two-sided unpaired t-test. **e** Bar plot showing the FANO factor number on strain-specific induced genes (2006 in BL/6 and 1205 in BALB/c mice). **f** Cluster visualization analysis (WGCNA) was performed on the expression of strain-specific induced genes (2006 in BL/6 and 1205 in BALB/c mice). Light color represents low overlap and progressively darker red color represents higher overlap. Blocks of darker colors along the diagonal are the modules. The gene numbers are labeled at the top of the figure.

## scRNA-seq analysis of BL/6 and BALB/c macrophages in the same tissue environment of chimeric F1 mice identifies cell intrinsic strain specific regulons

The epigenetic state and transcriptional responses of BL/6 and BALB/c TRMs to IL-4 stimulation in vivo could be influenced by strain specific homeostatic differences in the tissue environment prior to IL-4 exposure. To investigate whether cell intrinsic genetic differences between

BL/6 and BALB/c TRMs mediate IL-4 responses when they are generated within the identical tissue environment, we generated mixed bone marrow chimeric mice using CB6F1/J (F1) mice as recipients, reconstituted with mixed bone marrow (BM) from BALB/c and BL/6 mice (Fig. 6a). These mixed BM chimeric mice were then treated with IL-4 in vivo, and the total peritoneal exudate cells (PECs) were subjected to scRNA-seq. Based on the genetic variation between BL/6 and BALB/c

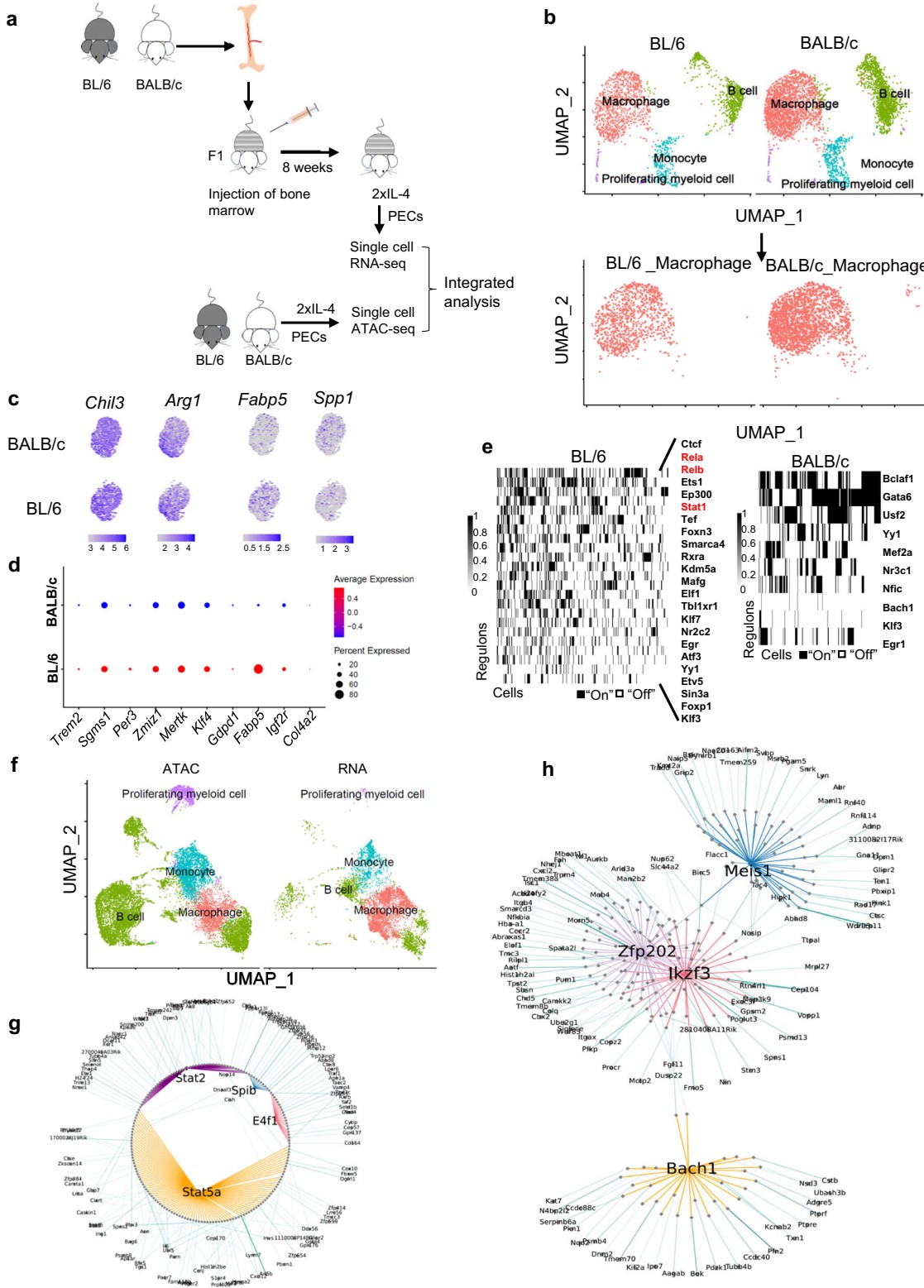

transcripts, we used Souporcell[34] to separate BL/6 and BALB/c PECs from chimeric F1 mice (Fig. 6b, upper). We subsequently focused on the macrophage subset (Fig. 6b, lower). As anticipated, *Chil3* and *Arg1* are expressed in both BL/6 and BALB/c TRMs, with expression observed in the majority of cells (Fig. 6c, left). In contrast, *Fabp5* demonstrates higher expression in BL/6 TRMs at both the population and average per-cell expression levels. Conversely, *Spp1* which is an important marker of tumor macrophages[35] exhibits higher population-level expression in BALB/c macrophages (Fig. 6c). We then validated

that the expression of IL-4-induced BL/6-specific genes, as indicated previously by bulk RNA-seq data, is also notably higher in BL/6 TRMs within the F1 mice. Notably, these genes include *Trem2*, *Sgmas1*, *Per3*, *Zmiz1*, and *Mertk*, whereby NF-κB motifs were identified in the promoter-distal regions through MEME-FIMO analysis, defined by chromatin loops from Hi-C data (Fig. 6d). Subsequently, we employed single-cell regulatory network inference and clustering (SCENIC) analysis to identify transcription factor (TF) regulons in BL/6 and BALB/c macrophages. This analysis revealed notable activity of NF-κB (Rela

**Fig. 6 | BL/6 and BALB/c cell intrinsic responses to IL-4 activation in the mixed bone marrow chimeras F1 mouse. a** Schematic illustration of the experimental protocol: Bone marrow (BM) cells from BL/6 and BALB/c mice were adoptively transferred into F1 mice. After 8 weeks, chimeric mice received two i.p. injections of 10 μg IL-4-Fc per mouse. Peritoneal cells were harvested 4 days after the first injection for scRNA-seq library construction. **b** UMAP plots visualized clusters of peritoneal cells derived from scRNA-seq of chimeric F1 mice. The clusters were color-coded based on their annotated cell names. The subset of macrophages was isolated for further analysis. **c** Feature plots show the selected genes expression of BL/6 and BALB/c macrophages. **d** Dot plots visualizes genes more expressed in BL/6 macrophages, of which *Trem2*, *Sgmas1*, *Per3*, *Zmiz1*, and *Mertk* have NF-κB motifs in the promoter-distal regions defined by chromatin loops from Hi-C data. Red indicates a higher average expression of the gene, and a larger dot size indicates that the gene is present in a larger percentage of cells within that cluster. **e** Regulons activities were computed for macrophages using SCENIC. The regulon names are labeled on the right side of the figure, with the red color-labeled names highlighting the SDTFs related regulons in BL/6 mice. **f** UMAP plots display the integration of scRNA-seq and scATAC-seq data. The subset of macrophages was isolated for further analysis. **g, h** SCENIC+ analysis was performed after integrating scRNA-seq and scATAC-seq datasets. Cytoscape visualization of separate eGRNs formed by Stat5a, Stat2, Spib, and E4f1 in BL/6 TRMs after IL-4 treatment (**g**), and the eGRN formed by Meis1, Bach2, Zfp202, and Ikzf3 in BALB/6 TRMs after IL-4 treatment (**h**).

and Relb) and STAT (Stat1) regulons in a distinct subset of BL/6 macrophages, which is not detected in BALB/c macrophages (Fig. 6e). Notably, only a distinct subset of BL/6 TRMs is regulated by the NF-κB. Hence, differences between strains could potentially be driven by just a small subset of the macrophage population. To investigate this subset further, we identified cells that have either the Rela or Relb regulon activity and performed a differential analysis with all the other macrophages (Supplementary Fig. 6a). Notably, Spp1 is more highly expressed in the subset with NF-κB regulon activity.

Next, we conducted single-cell ATAC-seq (scATAC-seq) on PECs from BL/6 and BALB/c mice that have been treated with IL-4 in vivo to examine the accessible chromatin landscape of peritoneal TRMs in BALB/c and BL/6 mice at single cell level. The macrophage populations were identified by integrating the scATAC-seq data with the scRNA-seq data from the mixed BM chimeric mice (Fig. 6f) and subsequently separated for analysis using Signac to identify strain-specific transcription factor (TF) motifs (Supplementary Fig. 6b). We found that bZIP motifs (e.g., FOS, JUNB) are more enriched in BALB/c macrophages (Supplementary Fig. 6b, right), consistent with ATAC-seq analysis of bulk cells, whereas KLF motifs are enriched in BL/6 macrophages (Supplementary Fig. 6b, left). Integrating analyses of chromatin accessibility and gene expression of individual cells can be used to infer gene regulatory networks (GRNs) through SCENIC+[36]. We employed SCENIC+ to construct GRNs for both BL/6 and BALB/c macrophages, after integrating the scATAC-seq data with scRNA-seq data and subsetting the macrophages from the total PEC population. The enhancer-driven Gene Regulatory Networks (eGRNs) constructed for IL-4-activated BL/6 and BALB/c TRMs reveal distinct transcription factor networks. SDTF motifs such as STATs and ETS motifs such as Spib (PU.1) and E4f1 exhibit higher activity in BL/6 TRMs, whereas homeobox TFs such as Meis1 are more active in BALB/c TRMs. These findings align with the data obtained from bulk ATAC-seq analysis (Fig. 6g, h).

In summary, these findings validate that the major genomic differences observed between bulk TRMs of BL/6 and BALB/c mice can be largely reproduced when analyzing macrophages isolated from the exact same tissue environment at a single-cell level. Therefore, cell-intrinsic variation significantly influences the strain-specific responses of peritoneal TRMs to IL-4 activation.

**Cell-intrinsic strain-specific responses to IL-4 and LPS synergy are observed in the same tissue environment of chimeric F1 mice**

To assess whether the extended synergy between IL-4 priming and LPS activation results in distinct functional consequences in BL/6 and BALB/c mice, we subjected IL-4-treated BL/6 and BALB/c mice to an in vivo challenge with a sub-lethal dose of LPS. High dose LPS treatment in vitro overwhelmed the transcriptional signature of IL-4 primed synergy in both strains. We used an adapted murine sepsis score (MSS) to evaluate the effects of sub-lethal dose LPS challenge (Fig. 7a). Consistent with a stronger transcriptional response to LPS in BALB/c peritoneal TRMs, we found that BALB/c mice have a higher MSS score than BL/6 mice after LPS treatment ($p = 0.0108$). However, IL-4 pre-treatment does not affect the MSS score in BALB/c mice after LPS

challenge, but for BL/6 mice IL-4 pre-treatment trended towards having a higher MSS score ($p = 0.0867$) (Fig. 7a).

To assess the cell intrinsic and strain specific in vivo transcriptional response to IL-4 and LPS synergy in the same tissue environment, we performed scRNA-seq analysis of PECs from mixed BM-chimeric mice (Fig. 7b, c). We compared naïve untreated mice (Ctrl), mice treated with LPS alone, and those pre-treated with IL-4 and subsequently challenged with LPS. The separation of BL/6 and BALB/c cells from the mixed BM-chimeric F1 mice was achieved through Souporcell[34] and cell types were annotated by singleR[37] (Fig. 7c). As expected, LPS treatment resulted in reduction of the macrophage population (Fig. 7d, Supplementary Fig. 6c) through the "macrophage disappearance reaction"[2]. LPS treatment also increased the neutrophil population in PECs (Fig. 7d, Supplementary Fig. 6c). IL-4 pre-treatment expanded peritoneal macrophages and reduced the loss of macrophages after LPS treatment (Fig. 7g and Supplementary Fig. 6e). There was sufficient number of macrophages in all groups to allow for meaningful downstream analysis. We identified genes that are differentially regulated between groups (logFC threshold ≥ 0.25). Consistent with bulk RNA-seq (Fig. 3b), we observed slightly more upregulation of genes by LPS in BALB/c macrophages compared to control naïve (Ctrl) macrophages, as opposed to BL/6 macrophages (Fig. 7e). However, in BL/6 macrophages, we identified a higher number of genes upregulated in the IL-4 + LPS treatment group compared to the LPS-only group, indicative of a more robust synergistic response on the BL/6 background, consistent with the bulk RNA-seq experiments (Fig. 3b).

Genes identified to be synergistically upregulated by IL-4 + LPS, such as *Ccl2*, *Ccl12*, *Ccl7*, and *Edn1*, had comparable percentage of cells expressing these genes between BL/6 and BALB/c macrophages. However, the level of expression per cell is higher in BL/6 macrophages compared with Ctrl (Fig. 7f, middle), consistent with a cell intrinsic response. Representative genes with no synergistic response (Fig. 7f, left), those exhibiting a synergistic response (Fig. 7f, middle), and those displaying a more pronounced response in the LPS treatment group in BALB/c macrophages compared to BL/6 macrophage (Fig.7f, right) are shown. Next, we re-clustered the macrophages into distinct phenotypes to identify greater granularity in macrophage heterogeneity (Fig. 7g) and applied SCENIC analysis to identify TF regulons associated with the distinct macrophage clusters (Fig. 7h) and different treatment groups (Fig. 7i). This analysis integrated data from both BL/6 and BALB/c mice, allowing for a comprehensive comparison of the clustering and treatment conditions. Some macrophage clusters experience significant depletion after LPS exposure, such as Cluster 2 and Cluster 4. By SCENIC analysis, the mostly depleted clusters are enriched with TRM-specific lineage and/or functional markers[38–40]. For example, Cluster 2 and Cluster 4 macrophages are distinguished by the activity of Gata6, Klf, and Fli1 regulons (Fig. 7h, Supplementary Fig. 6d). In Cluster 4, some macrophage activation-related transcription factors in TRMs are enriched, such as Foxp1 and Atf7[41]. Cluster 6 macrophages are more abundant in the IL-4 + LPS treatment group compared to the LPS-only group, particularly from the BL/6 background (Supplementary Fig. 6d). SCENIC analysis of Cluster 6 reveals

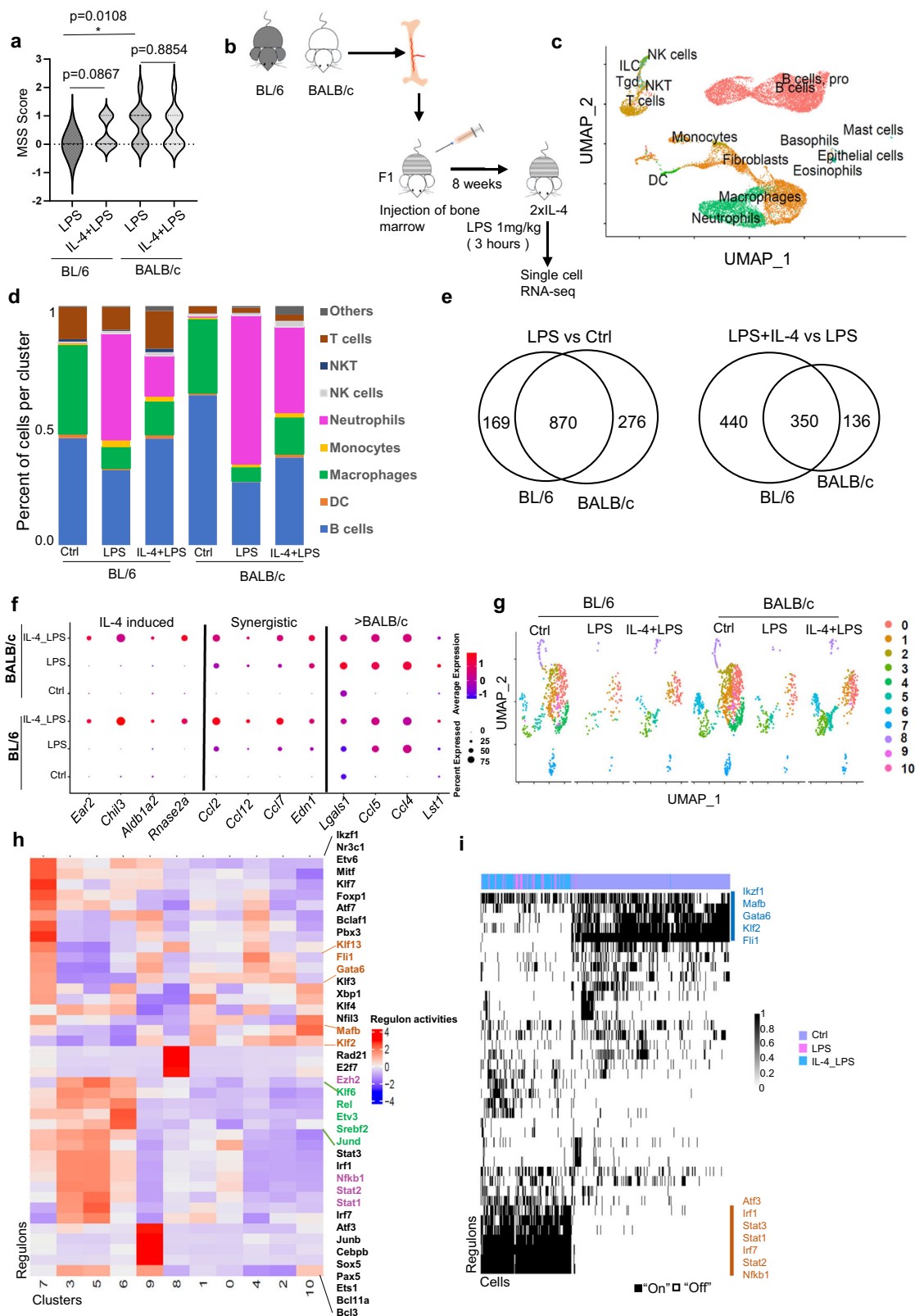

an enrichment of the NF-κB (Rel) regulon. Additionally, we found that clusters 3, 5, and 8 are also increased in the LPS + IL-4 group, especially in BL/6 mice. Among the motif analysis after SCENIC, we found that the Ezh2 regulon is more enriched in Cluster 8. Ezh2 has been shown to physically interact with RelA via the transactivation domain and can co-activate the transcription of a subset of NF-κB target genes with RelA[42]. Furthermore, we found that the Nfkb1 and Stats regulons are enriched

in clusters 3 and 5, which are also more induced in the LPS + IL-4 group in BL/6 mice. All clusters (3, 5, 6 and 8) increase in response to LPS + IL-4 treatment in BL/6, indicating NF-κB regulation. This is consistent with our conclusion that NF-κB is involved in a synergistic response in BL/6 mice. Gata6 regulon activity is observed in the majority of naïve peritoneal macrophages (Ctrl). However, after LPS treatment (including in the context of IL-4 pretreatment), the activity of SDTF regulons such as

**Fig. 7 | BL/6 and BALB/c cell intrinsic differences after in vivo IL-4 + LPS synergistic activation from mixed bone marrow chimeras F1 mice. a** Violin plot displays the effects of sub-lethal LPS treatment with or without prior IL-4 treatment on murine sepsis score (MSS). LPS (500 μg/kg) was administered after 4 days with or without prior IL-4-Fc injection, and the MSS score was observed the following day around 9:00 am ($n = 10$/group). $p$-values were calculated using a two-sided Mann-Whitney test. **b** Schematic illustration of the experimental protocol (see methods). **c** UMAP plot visualize clusters of peritoneal cells derived from bone marrow chimeric mice. The UMAP were generated from integrated samples, including control (Ctrl), LPS, and IL-4 + LPS treatment groups from both BL/6 and BALB/c strains. Cells are color-coded according to computationally determined cell clusters by the singleR program. **d** Histogram depicts the cell type composition as determined by single-cell analysis for each sample, showing a decrease in macrophage cluster (green) and an increase in neutrophil cluster (pink) after LPS treatment. **e** Significantly induced genes (logfc.threshold = 0.25) were identified (LPS vs Ctrl and LPS + IL-4 vs LPS). The left panel displays the overlap of genes significantly induced after LPS treatment compared to Ctrl in macrophages from both strains. The right panel displays the overlap of genes significantly induced after IL-4 + LPS treatment compared to LPS treatment alone in macrophages from both strains.

**f** The dot plot displays gene expression patterns, illustrating examples of no synergistic effect (left), synergistic effect (middle) following IL-4 + LPS treatment, and enhanced responses in BALB/c mice after LPS treatment (right). In the plot, red color represents a higher average expression of the gene, and a larger dot size indicates that the gene is present in a larger percentage of cells within that cluster. **g** The macrophage subset from the scRNA-seq data was re-integrated by selecting the "FindVariableFeatures" function with "nfeatures = 50" to visualize changes in cluster composition among samples. Cluster labels are represented as numbers on the right side of the figure. **h** Regulon activity was computed for macrophages from (**g**) using SCENIC analysis. The regulon names are labeled on the right side of the figure, with orange-colored names indicating the regulons in cluster 2 that disappear after LPS treatment, green-colored names indicate the regulons in cluster 6, whose population is increased after IL-4 + LPS treatment compared to LPS treatment in BL/6 mice, and purple-colored names indicate the regulons in cluster 3,5 and 8, whose population is increased after IL-4 + LPS treatment compared to LPS treatment in BL/6. **i** The regulon clustering from SCENIC analysis reveals that some regulons only contribute to Ctrl condition, labeled in blue, while another group of regulons exclusively contributes to the activation state induced either by LPS treatment alone or IL-4 + LPS treatment, labeled in orange.

Nkb1, Stats, and Irf becomes activated in the remaining TRMs (Fig. 7i). Overall, these data from BALB/c and BL/6 macrophages that are primed by IL-4 treatment prior to exposure to LPS in the same tissue environment of a chimeric F1 mice clearly indicates that the strain specific differences in IL-4 and LPS transcriptional synergy are driven by cell intrinsic effect.

To determine if the BL/6 and BALB/c alleles are epigenetically regulated differently in the same macrophage of a F1 mouse, we performed a deep ATAC-seq analyses on peritoneal macrophages from IL4-Fc treated F1 mice in comparison to control naïve untreated F1 mice. In order to confidently associate peaks with alleles, we restricted analyses to only reads that mapped perfectly to the genome and also overlap with a sequence variant (between BL/6 and BALB/c genomes), to determine if reads are from BL/6 or BALB/c alleles and to assess differences in regions of open chromatin unique to each parental strain (Supplementary Fig. 7). Of the 27,431 peaks of naive macrophages, 1115 are BL/6 specific and 1006 are BALB/c specific. Of the 27,619 peaks of IL-4-Fc treated macrophages, 1123 are BL/6 specific and 1106 are BALB/c specific (Supplementary Fig. 7a). Of the 30,927 merged peaks for Control and IL4 conditions, 1270 are induced on the BALB/c allele and 1300 induced on BL/6 allele by IL-4 (Supplementary Fig. 7b). We did not observe preferential enrichment of the NF-kB motif in the BL/6-specific IL-4 induced peaks, likely because there are only 623 peaks that do not also change in the BALB/c allele. However, we found that the homeobox motif, such as Meis1, is still more enriched in the BALB/c-specific IL-4 induced peaks, consistent with previous results. Additionally, we observed that the CTCF motif, which organizes genome structure[43], is more enriched BL/6-specific IL-4 induced peaks (Supplementary Fig. 7c, d). Hence, the majority of all peaks were cis regulated in both parental alleles and the observed differences between the parental strains could be due to differences in SDTFs protein abundance or activation, rather than sequence variation alone.

## Discussion

We found that natural genetic variation between BL/6 and BALB/c mice significantly influences epigenomic reprogramming of peritoneal TRMs in response to IL-4, leading to differences in their synergistic response to LPS activation. Integration of multiple stimuli by macrophages could be shaped by different genetic backgrounds and these results highlight the importance of studying transcriptional regulation across genetic backgrounds, rather than focusing solely on the BL/6 strain.

Our results align with recent studies on liver Kupffer cells, which showed preferential activation of AP-1 and MAF in BALB/c mice, while BL/6 Kupffer cells favored IRF, NF-κB, and LXR activity[44]. Similarly, we observed greater AP-1 activity in BALB/c TRMs, while BL/6 TRMs exhibited heightened NF-κB activity, which may explain the more pronounced IL-4/LPS synergy in BL/6 macrophages. Furthermore, IL-4 remodels the epigenome to expose NF-κB motifs in BL/6 BMDMs[18] as well as peritoneal TRMs, a process absent on the BALB/c background, driving the heightened synergistic response to LPS in BL/6 compared to BALB/c macrophages. In addition to confirming previously reported motif enrichment differences between BL/6 and BALB/c macrophages[44], our work demonstrated the functional consequence of this variation. IL-4 can remodel the BL/6 epigenome to expose NF-κB binding sites, contributing to a distinct and robust inflammatory response to LPS, akin to trained immunity. Trained immunity, where innate immune cells exhibit heightened responses to subsequent stimuli is important for how the immune system can adapt and enhance its responses over time[45–47]. How genetic background shapes immune responses may depend on how exposure to IL-4 could remodel epigenetic pathways in macrophages or monocytes[48], to increase capacity to respond to future challenges.

Natural genetic variations in type 2 immunity contributes to phenotypic diversity in inflammatory diseases, such as asthma and atopic dermatitis. Genetic differences in how immune cells integrate signals may also influence disease heterogeneity. While previous studies often focused on in vitro stimulated BMDMs[16], we examined large peritoneal cavity macrophages (LPM) that are F4/80^hiMHCII^low and depend on the expression of Gata6 for their differentiation[49,38]. In response to inflammation, LPM will aggregate at specific sites in the peritoneal cavity to contain infection or mediate tissue repair[50]. IL-4 treatment or infection with nematode parasites expands the LPM population through proliferation[51], while activating the macrophages to adopt a tissue repair phenotype, making this a useful in vivo model for studying type 2 macrophage activation across different genetic backgrounds.

In our genomic studies, we found strain-specific differences in TF motif enrichment, particularly for ETS (PU.1 or ELFs) and NF-κB in BL/6 peritoneal TRMs after IL-4 activation. These results may arise from the collaborative efforts of SDTFs with LDTFs, including PU.1, AP-1, and C/EBP (CCAAT/enhancer binding protein)[16]. Interestingly, although previous studies identified EGR2 as key to LPS synergy in BMDMs[18], we did not observe its enrichment in our TRMs, suggesting differences between macrophage subtypes. Our preliminary experiments in thioglycolate-induced monocyte-derived macrophages suggests that the activity of EGR2 is more significant in inflammatory macrophages.

NF-κB complexes are formed by members of the Rel family of transcription factors[21]. While LPS signaling typically involves RelA, the

specific subunit of the NF-κB complex driving IL-4-primed macrophages is unclear and different NF-κB subunits may competitively bind the same DNA sites[52]. Importantly, we found that variation in NF-κB motif sequences between BL/6 and BALB/c mice does not appear to contribute significantly to the differences in accessibility and transcription. However, binding of NF-κB can be dependent on LDTFs binding, as mutations in PU.1 or C/EBPβ motifs could abolish signal dependent binding of NF-κB[15]. Hence, the differences in NF-κB activity may not be driven by mutations in the cognate recognition motif but could be driven by genetic variation in the motifs of other LDTFs or SDTFs that collaborate to mediate the different response between BL/6 and BALB/c strains.

In addition to cell-intrinsic factors such as chromatin accessibility and 3D chromatin structure[53,54], the cytokine milieu of the tissue shapes macrophage phenotypes. Chromatin modifications are regulated by crosstalk between the environment and cell ontogeny through a small number of TFs[55]. The tissue environment regulates the expression and function of TFs that activate cis-regulatory enhancer elements[56–58]. Our work emphasizes that while macrophages are exposed to similar environmental cues, their transcriptional responses are shaped by their genetic background. Preliminary experiments indicate that IL-4 + LPS stimulated BL/6 macrophages may have an advantage after in vitro exposure to *Toxoplasma gondii* tachyzoites, but further experiments are required to draw firmer conclusions. In this manuscript we have focused on upregulated genes because the mechanisms of transcriptional activators (vs repressors) in relation to epigenetic changes is better understood. Future studies will investigate genes that are specifically downregulated by IL-4 in BL/6 and BALB/c macrophages.

We also utilized mixed bone marrow chimeric mice, allowing us to investigate BL/6 and BALB/c TRMs in the same peritoneal environment. Single cell sequencing confirmed that STATs, IRFs, and NF-κB regulons were enriched specifically in BL/6 TRMs, indicating that these differences are cell intrinsic and not influenced by the tissue microenvironment. It is possible that inherent differences in tonic IL-4 signaling between BALB/c and BL/6 mice may imprint epigenetic memory on the hematopoietic stem cells in the bone marrow, which were transferred into the mixed chimeric mice. One limitation of our studies is that we have not characterized epigenetic differences in the bone marrow between the BALB/c and BL/6 mice. Additionally, bone marrow transfers would also bring B cells from BL/6 and BALB/c mice into the same F1 environment, and we cannot rule out that this may affect macrophage intrinsic responses. An additional caveat of these studies is that we do not distinguish between embryonically and bone marrow derived resident peritoneal macrophages and have only data on male mice, while there are gender specific differences at the rate by which monocyte-derived F4/80hi macrophages displace the embryonic population with age[59].

Our study underscores the substantial impact of inherent genetic variations on the synergistic interplay between IL-4 stimulation and TLR engagement via LPS. Strain-specific epigenetic responses to IL-4 results in distinct transcriptional responses to subsequent TLR engagement. Hence, genetic variation may be a key influencer of how repeated activating signals affect epigenomic memory and synergistic responses. These insights contribute to our understanding of how genetic background of specific individuals influences inflammatory responses and hold potential to enhance our capacity to develop a personalized approach to therapy for inflammatory diseases driven by type-2 cytokines.

## Methods

### Mice and IL-4-Fc and LPS treatment
All experiments were performed under protocol number LPD16E, approved by the NIAID Animal Care and Use Committee. Male BALB/cJ and C57BL/6J mice, typically 6 to 8 weeks old, were purchased from

The Jackson Laboratory and/or bred in specific-pathogen-free facilities at the NIH. A fusion protein of mouse IL-4 with the Fc portion of IgG1 (custom order with Absolute Antibody) was generated to extend half-life with similar effects to IL-4−anti−IL-4 mAb complex. Fc fusion proteins (also known as Fc chimeric fusion proteins, Fc-Igs, Ig-based chimeric fusion proteins, and Fc-tag proteins) are composed of the Fc domain of IgG genetically linked to a peptide or protein of interest (mouse IL-4 in this project)[60]. Mice were injected i.p. with either PBS or 10 μg IL-4-Fc in 100 μl PBS on days 0 and 2, and peritoneal exudate cells (PECs) were harvested on day 4. PECs were enriched for >90% F4/80+ macrophages cells by using EasySep™ Mouse F4/80 Positive Selection Kit (100-0659, Stemcell Technologies). For *H. polygyrus* infection experiments, mice were infected with third stage (L3) *H. polygyrus* larvae by oral gavage. 200 Hp larvae was give per mouse. *H. polygyrus* were propagated as previously described[61]. Mice were sacrificed 13 days after *H. polygyrus* infection. In vivo, LPS (500 μg/kg) was administered after 4 days with or without prior IL-4-Fc injection, and the MSS score was observed the following day around 9:00 am, as described in Mai et al.[62]. *p*-values for the MSS score were calculated using a two-sided Mann-Whitney test. All experiments were performed under protocol number LPD16E, approved by the NIAID Animal Care and Use Committee.

### Generation of mixed BL/6 and BALB/c bone marrow chimera F1 mice
Mixed chimeric mice were generated by either irradiation or busulfan treatment. In one set of experiments, F1 mice: CB6F1/J (Jax no. 100007) were lethally irradiated with 950 rad and subsequently reconstituted i.v. with $2 \times 10^6$ bone marrow cell mixture 1:1 from BL/6 and BALB/c donor mice after depletion of T lymphocytes by negative selection on CD90.2 MACS columns (Miltenyi Biotec). Alternatively, F1 were treated i.p. with 25 mg/kg of Busulfan on day −6, −4, −2 before receiving on day 0 bone marrow cell mixture. All BM recipients were maintained on antibiotic TMS water containing Trimethoprim (0.13 mg/ml), Sulfadiazine (0.5 mg/ml), and 0.67 mg/ml Sulfamethoxazole (0.67 mg/ml) for 8 weeks. The chimerism of reconstituted mice was confirmed by FACS using APC anti-H-2Kb (clone AF6-885.5.3 Invitrogen) and Super bright 436 anti-H-2Kd (clone SF1-1.1.1 Invitrogen). Data were collected on BD Symphony and analyzed by FlowJo. The gating strategy used for FACS analysis of blood samples from bone marrow reconstituted F1 recipients is presented in Supplementary Fig. 6f. After 8 weeks, chimeric mice received two intraperitoneal injections of 10 μg IL-4-Fc per mouse 2 days apart. After 4 days of IL-4 treatment in vivo peritoneal cells were harvested for scRNA-seq. In other experiments, just before harvesting the peritoneal cells, LPS was injected at a dose of 1 mg/kg in vivo for 3 h. The peritoneal cells were then isolated from the chimeric mice and used to prepare the scRNA-seq library.

### ChIP and ChIP-seq library construction
ChIP experiments were performed as described previously[63]. Briefly, isolated macrophages were washed twice with PBS, pin samples for 10 min, 300 g at 4 °C, fixed at room temperature with 1% formaldehyde in PBS for 10 min. Reactions were quenched by adding glycine to a final concentration of 0.125 M, and cells were washed twice with cold PBS. Cells were resuspended in cell lysis buffer (1% SDS, 10 mM EDTA, 50 mM Tris-HCl pH8.1), incubated in ice for 1 h. The crosslinked chromatin was fragmented by sonication (Covaris, M220) to sheer size 200−2000 bp. The final sonication product was diluted 5 times by dilution buffer (0.01% SDS, 1.1% Triton X-100, 1.2 mM EDTA, 16.7 mM Tris-Hcl pH 8.0, 167 mM NaCl). ChIP was performed with 3 μg anti-H3K27ac (39133, Active Motif) antibody-bound Protein A beads (10001D, Invitrogen) and incubated at 4 °C overnight. Beads were collected by centrifugation, washed, and incubated at 65 °C for 4 h in elution buffer (50 mM Tris-HCl, pH 7.5; 10 mM EDTA; 1% SDS) to reverse cross-linking. ChIP DNA was purified by ChIP DNA Clean &

Concentrator (Zymo Research: D5205). For sequencing, ChIP-seq samples were pooled and sequenced on NextSeq2000 using TruSeq ChIP Library Prep Kit (IP-202-1012) and single-end sequencing. Two biological replicate ChIP-seq experiments were carried out. ChIP-seq data are available on the Gene Expression Omnibus (GEO) website (http://www.ncbi.nlm.nih.gov/geo/).

## Analysis of ChIP-seq data
The Real Time Analysis software (RTA 3.4.4) was used for processing raw data files, the Illumina bcl2fastq v2.20 was used to demultiplex and convert binary base calls and qualities to fastq format. Samples were trimmed for adapters using Cutadapt (1.18) before the alignment. The trimmed reads were aligned with mm10 reference using Bowtie2(2.2.6) alignment. Library complexity is measured by uniquely aligned reads using Picard (2.18.26)'s markduplicate utility. Peaks were called by MACS2(2.2.6) with –broad option on. HOMER (v4.10.4) findPeaks -style super was used to find super enhancers in ChIP-seq. Enrichment analysis of strain-specific enhancer regions based on biological process Gene Ontology (GO) terms was conducted using the Genomic Regions Enrichment of Annotations Tool (GREAT) (http://great.stanford.edu/). Bigwig files are generated by bamCoverage(deeptools/3.5.0), with the parameter −binSize 10 −normalize using RPGC.

## ATAC-seq library construction
The ATAC-seq library was created followed previously established procedures with some minor adjustments[64]. To start, 100,000 macrophage cells were subjected to lysis using 100 μl of Lysis buffer (containing 10 mM Tris-Cl at pH 7.4, 10 mM NaCl, 3 mM $MgCl_2$, 0.1% NP-40, and 0.1% Tween-20). Following a centrifugation at $500 \times g$ for 10 min at 4 °C, the resulting nuclei were reconstituted with 50 μl of a transposition mixture (FC-121-1030, Illumina; comprising 1X Tagment DNA buffer, Tn5 Transposase, and nuclease-free $H_2O$), and this mixture was then incubated for 30 minutes at 37 °C in a thermomixer. The transposed DNA was subsequently purified using MinElute columns (28004, QIAGEN) and then amplified using Nextera sequencing primers and the NEB high-fidelity 2X PCR master mix for 11 cycles (M0541, New England Biolabs). PCR-amplified DNA was purified using MinElute columns (28004, QIAGEN) and ATAC-seq samples were pooled and sequenced on NextSeq 2000 P2 using paired-end sequencing.

## Analysis of ATAC-seq data
The Real Time Analysis software (RTA 3.9.25) was used for processing raw data files, the Illumina bcl2fastq v2.17 was used to demultiplex and convert binary base calls and qualities to fastq format. Samples were trimmed for adapters using Cutadapt (1.18) before the alignment. The trimmed reads were aligned with mm10 reference using Bowtie2(2.2.6) alignment. Library complexity is measured by uniquely aligned reads using Picard (2.18.26)'s markduplicate utility. The peaks were called using Genrich (0.6.1) (genrich -t sample_sortedByRead.bam -o sample.narroPeak -j -y -r -v -d 150 -m 5 -e chrM,chrY -E blacklist_regions.bed -f sample.bdg -b sample.bed). Differentially accessible peaks after IL-4 treatment were analyzed by DiffBind (2.10)(http://bioconductor.org/packages/release/bioc/vignettes/DiffBind/inst/doc/DiffBind.pdf), with FDR ≤ 0.001 define induced peaks at different time point compared with Ctrl. HOMER (v4.10.4) findMotifsGenome.pl was used to investigate the motif enrichment of peaks region selected. chromVAR[23] was used to find the motif enriched score in all ATAC-seq samples. Bedtools (2.30.0) intersect option are used to find the overlap of the ATAC-seq and H3K27ac ChIP-seq. Co-expression networks were constructed using the WGCNA (Weighted gene coexpression network analysis) algorithm implemented in R WGCNA package[33]. Raw expression values of selected differential changed genes were normalized with variance StabilizingTransformation (vst) function in R software. To obtain co-expressed modules, the parameters were adjusted to minModuleSize = 30 to cut the tree. For genome tracks, bigwig files for heatmap production were created from bam files using deepTools with the parameter−binSize 10 −normalize using RPGC. Heatmaps displaying normalized read densities of ATAC-seq and ChIP-seq samples were generated with the computeMatrix and plotHeatmap modules of the deepTools package[65]. Genome tracks were explored and visualized using the IGV browser.

## Analysis of F1 ATAC-seq data
Data was mapped to C57BL/6J and BALB/cJ genomes respectively and then shifted to mm10 coordinates using MMARGE[7]. To analyze from which parental allele each read comes, only perfectly mapped reads that overlap a mutation between C57BL/6J and BALB/cJ were considered. Then peaks that were unique to C57BL/6 and BALB/cJ parental genomes were annotated with all perfectly aligned, mutation overlapped reads. To assess globally the amount of cis- and trans-regulated regions of open chromatin, HOMER was used to call peaks with findPeaks.pl with default parameters on all perfectly aligned reads. Then, peaks were annotated with all perfectly aligned reads that spanned mutations.

## RNA isolation and RNA-seq library construction
Total RNA was isolated using the RNeasy Plus Mini Kit (74134, QIAGEN). mRNA-seq samples were pooled and sequenced on NextSeq 2000 P2 using Illumina® Stranded mRNA Prep and paired-end sequencing. Three independent replicates were used for RNA-seq analysis.

## RNA-seq analysis
Reads of the samples were trimmed for adapters and low-quality bases using Cutadapt (1.18) before alignment with the reference genome (mm10) and the annotated transcripts using STAR (2.6.0c). The mapping statistics are calculated using Picard software (2.18.26). Library complexity is measured in terms of unique fragments in the mapped reads using Picard (2.18.26)'s MarkDuplicate utility. In addition, the gene expression quantification analysis was performed for all samples using STAR/RSEM (1.3.2) tools. EBSeq (1.24.0) was used to produce normalized reads for all samples and to identify differential RNA expression in BL/6 and BALB/c samples compared with Ctrl or LPS. To see the significantly trend different after IL-4 treatment or LPS treatment or IL-4 + LPS treatment compared with Ctrl between BALB/c and BL/6 mice, we used DESeq2[66]. Differentially expressed genes were identified using DESeq2 following the model:

ddsTC <- DESeqDataSetFromMatrix(countData = countData, colData = colData, design = ~Strain+Treat+Strain:Treat; ddsTC <- DESeq(ddsTC, test = "LRT", reduced = ~ Strain+Treat).

All differentially changed genes were selected based on an FDR ≤ 0.05 after DESeq2 analysis; both genes were selected that were more or less than 2-fold up for further analysis. *k-means* analysis of RNA expression data was carried out in MATLAB using fold-change compared with Ctrl for IL-4 treat samples in Fig. 1f, fold-change compared with LPS treatment in Fig. 3c, and fold-change compared with Ctrl in Supplementary Fig. 3c, with correlation as the distance metric, repeat clustering set to 5, and other parameters set to default. The programs findMotifs.pl were used to identify transcription factor binding motifs within selected promoter regions (−400 bp to +100 bp from TSS). For visualization, bigwig files were generated by deepTools with the parameter –binSize 10 –normalizeUsing RPGC. Genome tracks were explored using the integrative genomics viewer (IGV) browser.

## Hi-C libraries and sequencing
Chromatin interaction (Hi-C) libraries preparations were performed by Arima Genomics (https://arimagenomics.com/) using the Arima-HiC kit that uses two enzymes (P/N: A510008). The resulting Arima-HiC proximally ligated DNA was then sheared, size-selected around 200–600 bp using SPRI beads, and enriched for biotin-labeled

proximity-ligated DNA using streptavidin beads. From these fragments, Illumina-compatible libraries were generated using the KAPA Hyper Prep kit (P/N: KK8504). The resulting libraries were PCR amplified and purified with SPRI beads. The quality of the final libraries was checked with qPCR and TapeStation Systems. HiC-seq samples were pooled and sequenced on NovaSeq and paired-end sequencing.

## Hi-C data analysis

All Hi-C data were processed with Juicer (v.1.6[29]) pipelines with default settings. Sites for Phase Genomics cutting enzyme (GATC) were detected using the generate_site_positions.py, Sites for Arima Genomics cutting enzyme (^GATC, G^ANTC) were obtained from [Arima restriction enzyme files. Accessed 9 April 2020.ftp://ftp-arimagenomics.sdsc.edu/pub/HiCPro_GENOME_FRAGMENT_FILES]. Juicebox (2.3.0) was used to visualize the Hi-C heatmap. We defined TAD boundaries as regions 100 kb upstream of the TAD start and 100 kb downstream of the TAD end. Strain-specific loops were identified using Bedtools (2.30.0). Subsequently, we located promoter-based loops and annotated the promoter-distal regions using the HOMER (v4.10.4). Next, we overlapped the promoter distal regions with strain-specific highly expressed genes, as defined by DESeq2 analysis, for motif analysis in Fig. 4a. We employed MEME-FIMO (5.4.1) to scan for the locations of strain-specific active transcription factors (TFs) within promoter-distal regions (enhancers) that are linked to strain-specific loops associated with highly expressed genes. The presence of motif sites within the enhancer region indicates a regulatory relationship between the TF and the gene.

## scRNA-seq libraries and sequencing

Using a Chromium Next GEM Single Cell 3′ Kit v3.1, 4 rxns (1000269, 10X Genomics), Chromium Next GEM Single Cell 3′ Gel Bead Kit v3.1, 16 rxns (PN-1000122) and Chromium Next GEM Chip G Single Cell Kit (PN-1000120, 10X Genomics), the peritoneal cells suspensions in PBS, at 2000 cells per ul, were loaded onto a Chromium single cell controller (10X Genomics) to generate single-cell Gel beads-in-emulsion (GEMs) according to the manufacturer's protocol. Briefly, approximately 10,000 cells per sample were added to a chip to create GEMs. Cells were lysed and the bead captured poly(A) RNA was barcoded during reverse transcription in Thermo Fisher Veriti 96-well thermal cycler at 53 °C for 45 min, followed by 85 °C for 5 min. cDNA was generated and amplified. Quality control and quantification of the cDNA was conducted using Agilent Genomic DNA ScreenTape Analysis kit (5067-5366 for Genomic DNA Reagents and 5067-5365 for Genomic DNA ScreenTape) in the TapeStation system. scRNA-seq libraries were constructed using a Chromium Next GEM Single Cell 3′ Library Kit v3.1 (PN-1000158, 10X Genomics) and Single Index Kit T Set A, 96 rxns (PN-1000213, 10X Genomics). 10x Genomics Single Cell Gene Expression libraries were sequenced on a NextSeq 2000 run. The sequencing run was setup as a 28 cycles + 90 cycles non-symmetric run. Demultiplexing was done allowing 1 mismatch in the barcodes.

## scRNA-seq data processing

The analysis was performed with the Cell Ranger 7.1.0 software using the default parameters. The number of cells captured ranges from 4862 to 9060 and mean reads per cell ranges from 18,629 to 52,533. Cells with extremely low number of UMI counts were filtered out. The results from Cell Ranger were processed in R v3.6.3 with Seurat v3.2.3[67] using default parameters unless otherwise specified. Souporcell (https://github.com/wheaton5/souporcell) was used to separate the cells from CB6F1/J (Jax no. 100007) into BL/6 and BALB/c cells with singularity exec function. Quality control filtering was applied to each sample to eliminate downstream analysis of empty droplets, low-quality cells, and potential doublets. Then, we used the Seurat (v3.2.3)

package to perform calculation of the number of unique genes detected in each cell (nFeature_RNA), the total number of molecules detected within a cell (nCount_RNA), and the percentage of reads that map to the mitochondrial genome. The differential peaks in Fig. 7e were identified using the FindMarkers function in Seurat (v3.2.3) with the following parameters: logfc.threshold = 0.25, only.pos = TRUE, test.use = "MAST". To identify the potential key transcriptional regulators after IL-4 treatment, we used pySCENIC (v0.10.)[68] to explore the potential transcriptional regulators.

## Multiome-seq assay (snRNA-seq and snATAC-seq) from peritoneal cells

Peritoneal cells were isolated from both BL/6 and BALB/c mice after IL-4 treatment. Nuclei isolation for snRNA-seq and snATAC-seq was optimized and performed following the demonstrated protocol CG000365 from 10x Genomics. The libraries were prepared following the 10x Genomics user guide CG000338 for Single Cell Multiome ATAC + Gene Expression Reagent Bundle (PN-1000285,10xGenomics). Only the ATAC Library Construction is performed by Sample Index Plate N, Set A (3000427, 10xGenomics). snRNA-seq libraries failed QC and were not sequenced. 10x Genomics Single Cell ATAC libraries were sequenced on a NovaSeq 6000 run.

## scATAC-seq data analysis

The analysis was performed with the Cell Ranger ATAC 2.0.0 software using the default parameters. Quality control of scATAC-seq data from BL/6 and BALB/c strains was performed in R (v.4.3.0) using Signac (v.1.1.0)[69]. Rliger (1.0.0)[70] was used to integrate scATAC-seq and scRNA-seq data. Parameters included k = 20 for the optimizeALS function, n_neighbors=10, min_dist=0.3, and cosine distance for the runUMAP function. A resolution of 0.3 was used for the Louvain community detection. Cell types were labeled by cross-referencing the integrated scATAC-seq dataset with scRNA-seq datasets. Cell labels were transferred using the Seurat (4.9.9) package, focusing on the macrophage subset for downstream analyses.

Enhancer gene regulatory networks (eGRNs) were identified using SCENIC+[36], an algorithm utilizing paired single-cell transcriptomic and open chromatin data. In our analyses, we used a search space of 150 kb for inferring region-gene relationships and retained regulons with a minimum of 10 target genes. For pycisTopic (1.0.3), we executed the essential steps of the SCENIC+ workflow, including object creation, topic modeling, dimensionality reduction, dropout imputation, and DAR (Differential Accessibility Region) inference using default parameters. The topic modeling process employed serial LDA (Latent Dirichlet Allocation) with Collapsed Gibbs Sampler, with 500 iterations. The generated topics spanned from two to 48, and the final model selection yielded 10 topics for BALB/c and 6 topics for BL/6.PycisTarget (1.0.3) was run using default parameters, incorporating cisTarget and differential enrichment of motif (DEM). The analysis used the bulk consensus peaks motif databases. SCENIC+ was run with default parameters using http://sep2019.archive.ensembl.org/ as the BioMart host. High quality regulons were selected based on the correlation between gene-based regulon AUC and region-based regulon AUC (>0.7).

## Statistical analysis

In Fig. 7a, the p-values were calculated using a two-sided Mann-Whitney test. In others figures, p-values were calculated using an unpaired t-test, two-sided. Data are presented as mean ± standard deviation. Gene expression comparisons were reported by adjusted p values (i.e., FDR) from DESeq2[66].

## Reporting summary

Further information on research design is available in the Nature Portfolio Reporting Summary linked to this article.

## Data availability

ChIP-seq, ATAC-seq, RNA-seq, Hi-C, sc-RNA-seq data, scATAC-seq were deposited into the Gene Expression Omnibus database under accession number GSE248038 and are available at the following URL: https://www.ncbi.nlm.nih.gov/geo/query/acc.cgi?acc=GSE248038. Source data are provided with this paper.

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

## Acknowledgements

We thank all the members of Loke Lab in Laboratory of Parasitic Diseases (LPD) for valuable discussion. We also thank Dr. Iain Fraser and Dr. Verena M. Link for critical comments on the manuscript. This research was supported by the Division of Intramural Research, National Institute of Allergy and Infectious Diseases, National Institutes of Health (NIH).

## Author contributions

M.Z. and P.L. designed research; D.J generated mice; M.Z. performed ATAC-seq, ChIP-seq, bulk RNA-seq, scRNA-seq, scATAC-seq, and Hi-C; M.Z. analyzed data; J.B.L. performed Soupercell analysis and K.M.H. performed SCENIC plus analysis. V.M.L. and Y.B. performed mutation related analysis and helped to edit the manuscript. O.O. helped with the animal experiment. C.O.S.S. helped with the animal experiment and also assisted in editing the manuscript. M.Z. and P.L. prepared the manuscript.

## Funding

## Competing interests

The authors declare no competing interests.
