## [Peer Review file · Nature Communications]

Genetic variation in IL-4 activated tissue resident macrophages determines strain-specific synergistic responses to LPS epigenetically

Corresponding Author: Dr Png Loke

Version 0:

Reviewer comments:

Reviewer #1

(Remarks to the Author)

Zhao et al. perform a deep multimodal analysis of the transcriptional and epigenetic response of tissue resident murine peritoneal macrophages to IL-4 in vivo in two strains of mice. They further explore the synergistic response of these macrophages to the combination of IL-4 and LPS. This manuscript is especially noteworthy for this analysis being performed in vivo in tissue resident macrophages, rather than in vitro in bone marrow-derived macrophages, which is much more common, and which comes with many caveats. The results are robust, although not all that unexpected, in that the two inbred strains of mice show only partially overlapping responses to the different stimuli, and that the magnitude of their synergistic response to IL-4/LPS is also different. Nevertheless, this represents an important proof of concept that genetic variation plays a large role in how tissue resident macrophages respond to diverse stimuli both in isolation and when encountered simultaneously. The experiments performed in mixed bone marrow chimeras are excellent in showing that the different transcriptional responses and different synergistic responses are cell intrinsic to the macrophages.

Major points:

1) Could the authors compare the responses in the two strains of mice between treatment with IL-4-Fc and infection with *H. polygyrus*? What is the overlap of genes, etc? Are there genes uniquely induced with IL-4-Fc or with *H. polygyrus* infection? How similar are these responses? Am I correct in my reading of the data that in BL/6 mice, only IL-4-Fc allowed the RELB (NFkB) motif to be found in the promoter region of the IL-4-induced genes, while only in *H. polygyrus* infection was the IRF motif found in the promoter of the infection-induced genes? My understanding is that in the IL-4-Fc system, ATAC-seq allowed both of these motifs to be found in accessible genes in BL/6 macrophages.

2) I was intrigued by the finding in the scRNA-seq data that in BL/6 TRMs, only a distinct subset of the TRMs appeared to be regulated by NFkB (Figure 6E). As the authors mention, this implies that "differences in strains could potentially be driven by just a small subset of the macrophage population." Is there a way to look specifically at those cells which show the NFkB regulome, and see how these cells differ from the other macrophages transcriptionally (in terms of which genes they express more or less of)? For example, could these cells be highlighted on a UMAP plot, and would they represent a distinct region of the BL/6 macrophage cluster?

3) Could the authors speculate in the discussion on the durability of the epigenetic changes induced by IL-4 receptor signaling in the TRMs? For example, how long after the IL-4-Fc treatment is the synergistic response still seen? Are these chromatin accessibility changes induced by IL-4 receptor signaling long lasting, such that exposures to LPS at a much later time point would also show the synergistic response?

4) What would the transcriptional and epigenetic responses to IL-4 or IL-4/LPS look like in CB6F1 mice? Would changes in chromatin accessibility after IL-4-Fc happen differently on the BL/6 allele vs. the Balb/c allele for any given gene?

Minor points:

1) Can the authors point to another reference where they have used the IL-4-Fc fusion protein to describe how it was

generated in a bit more detail, or describe it here more completely.

2) Please add the methodology for the *H. polygyrus* infection experiments. At what day of infection are the mice sacrificed for purification of the peritoneal macrophages? What dose of L3 larvae was given, etc.

3) Line 511 of the Discussion has a sentence with what may be an error: "Notably, BMDM experiments also identified EGR2 except NF- κ B as a key...". Not sure what is meant here, I assume the "EGR2 except" is to be deleted?

4) The font in the legends of Fig S3D and Fig S4C should be changed to match the rest of the figures.

Reviewer #2

(Remarks to the Author)

The manuscript offers a thorough examination of the strain-specific response of tissue-resident memory T cells (TRM) to IL-4 and its synergistic effects with subsequent LPS stimulation. The integration of multiomic approaches, including bulk RNAseq, ATACseq, ChIPseq, and scRNAseq, enriches the analysis and provides valuable insights.

To enhance the manuscript's quality further, I recommend addressing the following points:

1. Given BALB/c mice's known propensity for stronger type 2 responses compared to BL/6, it's crucial to investigate whether inherent differences in tonic endogenous IL-4 levels exist between the strains. Such disparities may imprint epigenetic memory on TRM or hematopoietic stem cells (HSC) in the bone marrow, even though bone marrow chimeras are utilized to address cell-intrinsic differences.

2. The assertion that BL/6-specific genes are expressed at higher levels than BALB/c-specific genes, as depicted in FigS1A, might be overstated. Visual assessment of the normalized heatmap expression alone may not suffice to discern these differences accurately (lines 110-111).

3. In FigS2D, the de novo motif analysis should ideally yield multiple motifs instead of just one. Providing the entire motif list and justifying the selection of specific motifs for demonstration would enhance clarity and transparency.

4. Besides examining chemokine genes in Fig3E, the authors should also assess proinflammatory genes such as IL-1, TNF, and IL6, given IL-4's reported synergistic upregulation of immune response, as indicated in the C2 GO analysis (FigS3E). Similarly, including the full motif list as supplementary data for the de novo analysis in Fig3G would provide a more comprehensive understanding.

5. The adapted murine Speis score (MSS) requires definition, and detailed scoring criteria should be included in the materials and methods section to facilitate assessment of its validity.

6. In Fig7D, presenting actual cell counts in addition to percentages would offer more informative insights, especially considering the potential surge in neutrophil counts following LPS treatment. It's also essential to examine basal cell counts in the IL-4 treated group without LPS treatment to ascertain the impact of IL-4 on peritoneal macrophage expansion.

7. The interpretation of macrophage heterogeneity and clustering, along with associated regulons, requires clarification. For instance, while both C2 and C4 appear depleted after LPS exposure, only C2 is discussed. Similarly, although C6 and C9 seem to increase in the IL-4+LPS group, only C6 is addressed.

8. Regarding the discussion on strain-specific effects in regulons (Fig7H and Fig7I), it appears that some information may be lost, or combined data rather than strain-specific data may have been utilized for the analysis. Providing details about the analysis methodology would help clarify this aspect.

Minor comments:

1. There is a punctuation error in line 44.

2. Recent reports on IL-4-induced trained immunity (PMID: 37291433, 37149865, 36173104) are relevant and should be included for discussion.

Reviewer #3

(Remarks to the Author)

Zhao and colleagues present a significant collection of data indicating strain-specific difference in tissue resident macrophages in response to IL-4 mediated activation. Their data clearly show in vivo IL-4-Fc causes significant strain specific transcriptomic changes in BL/6 and Balb/C mice, with a greater effect in BL/6 mice. *H. polygyrus* confirms an alternative macrophage activation result that is greater in BL/6 mice cf. Balb/c. IL-4 increased inducer in BL/6 greater than Balb/c. In Balb/c IL-4 barely affects LPS signaling, yet it change BL/6 responses. The work is of significance to the field and supports varying immune "prone" responses particularly in Balb/c and C57Bl/6 mice. However, there remains some importance questions that if answered could potentially strengthen this manuscript.

Major comments:

1. Were the upregulated genes in response to IL-4 stimulation in TRMs and BMDMs similar? This was not discussed and may provide clarity on conserved responses across species and cellular compartments/subtypes.

2. The authors clearly demonstrated that IL-4-upregulated genes are greater in BL/6/. However, GO terms are not very informative on function (Figure S1B). While greater transcriptional function is reported, what is the function? This should be discussed in better detail to extrapolate function.

3. The biggest weakness is that the sole focus on macrophage epigenome, yet the treatment with IL-4 or LPS is in vivo. How are the changes in macrophage function not due to the influence of other cells namely T cells/B cells responding to IL-4, that then drive the macrophage response?

4. Is the responsiveness to IL-4 augmented in macrophages from Bl/6 mice because they are more Th1 prone and likely

receiving basal/tonic Th1 cytokine stimulation as opposed to balb/c mice that may be desensitised to IL-4 stimulation? The transcriptome while important and predictive requires validation. Is the function of the macrophages from each species different in vitro in response to IL-4 +/- LPS?

Reviewer #4

(Remarks to the Author)

The authors showed through a comprehensive set of bulk RNA and single cell RNA experiments that TRMs from BL/6 mice have a more pronounced response to IL-4 than TRMs from BALB/c mice. Through computational predictive analyses, 1-dimensional and 3-dimensional chromatin analyses the authors showed the more pronounced response in BL/6 mice compared to BALB/c mice arose from epigenomic remodelling mediated by NF- κ B binding sites that were intrinsic to the cells, irrespective of the tissue environment the cells were found in. Lastly, the authors showed the synergistic response of IL-4 and LPS treatment could also be attributed to epigenomic remodelling mediated by NF- κ B.

Comments:

1. Lines 111-112: "Gene ontology (GO) analysis of BL/6 or BALB/c specific genes indicates some genotype specific functional differences in chemotaxis and cell migration (Fig S1B)." – However, the GO terms in figure S1B for both strains are related to cell locomotion, taxis and migration. Since GO terms are hierarchical and can belong to the same "family" of terms despite sometimes different wording, could the author please explain/elaborate on these genotype specific differences?
2. Throughout the text, the authors focus on upregulated genes, despite evidence from their figures also showing a number of downregulated genes, including the identification of motifs for those genes. Can the authors please explain why they have chosen to focus only on upregulated genes?
3. In the text and figures, the authors said they infected BL/6 and BALB/C mice with *Heligomosomoides polygyrus* (*H. polygyrus*), however, the methodology used for these experiments have not been included in the methods section of the manuscript.
4. Lines 156-159: "Notably, BALB/c macrophages have more strain specific constitutive regions (n = 18,336 in BALB/c versus n = 3,731 in BL/6), whereas BL/6 macrophages have more IL-4 inducible accessible regions (n = 4,801 in BL/6 versus n = 3,660 in BALB/c) (Fig. 2A and 2B)." The difference in strain specific constitutively active regions is quite pronounced between the strains, whereas the differences following IL-4 stimulation are much less so. Are these differences of ~1200 regions significant?
5. In figure S3B, there are no labels for which GO terms are from BALBc and which are from BL/6 mice.
6. Lines 381-382: "Notably, only a distinct subset of BL/6 TRMs is regulated by the NF- κ B. Hence, differences between strains could potentially be driven by just a small subset of the macrophage population." It isn't until the end of the following results section (lines 444-457) that the authors expand on this point leaving the reader unsure of whether they will find out more about this specific subset.
7. Lines 411-412: "We used an adapted murine sepsis score (MSS) to evaluate the effects of sub-lethal dose LPS challenge (Fig. 7A)." Please provide more information in the methods on how the MSS score was adapted for future reproducibility.
8. For the majority of the figures, the authors have presented BL/6 data before BALB/c data, however there are some inconsistencies such as in Fig 7.A and 7.E where BALB/c data is presented before BL/6 data. Please fix.
9. Lines 428-429: "To identify differential peaks among groups, our focus was exclusively on highly induced genes (logFC threshold ≥ 0.25)". Did the authors possibly mean a log₂ fold change greater than or equal to 0.25 which would translate to a fold change of 2 or higher?
10. Discussion feels scattered with paragraphs jumping a lot and in certain instances displaying conversational language. There are also a lot of acronyms, with no list of abbreviations which reduce the readability of the text. Additionally, in several places, sentence structures are awkward and could be revisited to improve readability.
11. Authors need to justify the use of only male mice in their work.

Version 1:

Reviewer comments:

Reviewer #1

(Remarks to the Author)

In this revision the authors have significantly improved the manuscript. In particular I appreciate the new experiment testing the durability of the response to IL-4-Fc and the experiment in the CB6F1 mice. I also appreciate the comments of the other Reviewers, which have also improved the manuscript. I have only two minor comments based on two of the new

supplementary figures.

1) In new Fig S1B (in response to a critique from Reviewer 2), the text now reads "Quantified read counts indicate that BL/6-specific genes are expressed at significantly higher levels compared to BALB/c-specific genes (Fig. S1B). Could the authors add the information on the statistical test used to make this claim, and show p values in some way on the figure (perhaps as asterisks) for the relevant comparisons made.

2) In new Fig S1D, in response to my critique, I think the x axis labels are missing some detail. Are the different columns supposed to represent Ctrl and IL-4 or Ctrl and H. polygyrus, in both BL/6 and Balb/c?

Reviewer #2

(Remarks to the Author)

I acknowledge the efforts of the authors to improve the manuscript and their responses to my previous questions. I have no further questions, and the manuscript appears to be in good shape.

Reviewer #3

(Remarks to the Author)

The authors have addressed or attempted to address my concerns with relevant additional data and experiments that may have been inconclusive.

I have no further comments.

Reviewer #4

(Remarks to the Author)

Thank you to the authors for answering the reviewer questions thoughtfully and thoroughly. The manuscript is much improved and the findings quite interesting.

We appreciate the constructive comments and suggestions from the reviewers and the editors, which have substantially improved our revised manuscript. Additional experiments and analyses have been performed and provide a substantial amount of new information and additional clarification for the study. These changes have been highlighted in the revised manuscript. We believe we have addressed most of the reviewers' concerns to the best of our abilities during a reasonable time frame of the revision.

Importantly, we performed additional ATAC-seq experiments on peritoneal macrophages to examine the durability of the epigenetic changes and found surprisingly that after approximately 2 weeks the epigenetic changes have reverted to control conditions. This indicates that the synergistic response to LPS stimulation is not long lasting.

Additionally, we performed additional ATAC-seq experiments on F1 mice to examine IL-4 mediated chromatin remodeling on BL/6 and BALB/c alleles in the same cell. Our results show that while there are allele specific changes in chromatin accessibility that are happening differently on BL/6 and BALB/c alleles of certain genes, the majority of chromatin changes are conserved, indicating that the cell intrinsic abundances of major TFs within the F1 cells could be more important than allelic differences in sequence variation.

Overall, these experiments, as well as the additional analyses described below have significantly improved the manuscript. *A point by point response is included below.*

Reviewer 1:

Zhao et al. perform a deep multimodal analysis of the transcriptional and epigenetic response of tissue resident murine peritoneal macrophages to IL-4 in vivo in two strains of mice. They further explore the synergistic response of these macrophages to the combination of IL-4 and LPS. This manuscript is especially noteworthy for this analysis being performed in vivo in tissue resident macrophages, rather than in vitro in bone marrow-derived macrophages, which is much more common, and which comes with many caveats. The results are robust, although not all that unexpected, in that the two inbred strains of mice show only partially overlapping responses to the different stimuli, and that the magnitude of their synergistic response to IL-4/LPS is also different. Nevertheless, this represents an important proof of concept that genetic variation plays a large role in how tissue resident macrophages respond to diverse stimuli both in isolation and when encountered simultaneously. The experiments performed in mixed bone marrow chimeras are excellent in showing that the different transcriptional responses and different synergistic responses are cell intrinsic to the macrophages.

Response to Reviewer

We thank the Reviewer for the enthusiasm for our approach of focusing on tissue resident macrophages instead of bone marrow derived macrophages, as well as our usage of mixed bone marrow chimeras to show cell intrinsic responses of the macrophages. We also appreciate the endorsement for the robustness of the results and the importance of the proof of concept that genetic variation influences signal integration in macrophages.

We note that the most unexpected findings from this study are that the IL-4-induced epigenetic reprogramming of peritoneal macrophages to become hyperresponsiveness

to LPS stimulation for certain genes is strain specific on the C57BL/6 background and less so on the BALB/c background. This emphasizes the importance of transcriptional regulation studies on different genetic backgrounds.

Major points:

1) Could the authors compare the responses in the two strains of mice between treatment with IL-4-Fc and infection with *H. polygyrus*? What is the overlap of genes, etc? Are there genes uniquely induced with IL-4-Fc or with *H. polygyrus* infection? How similar are these responses? Am I correct in my reading of the data that in BL/6 mice, only IL-4-Fc allowed the RELB (NFκB) motif to be found in the promoter region of the IL-4-induced genes, while only in *H. polygyrus* infection was the IRF motif found in the promoter of the infection-induced genes? My understanding is that in the IL-4-Fc system, ATAC-seq allowed both of these motifs to be found in accessible genes in BL/6 macrophages.

We have now compared the transcriptional response of IL-4-Fc and *H. polygyrus* and found that there is some overlap of significantly induced genes, but there are more genes that are uniquely induced by *H. polygyrus* infection, in both BL/6 and BALB/c strains. These results are now included in Fig. S1. *H. polygyrus* induces macrophages that are phenotypically more different transcriptionally than naïve peritoneal macrophages likely because of a more complex stimuli of the entire parasite response compared to a single cytokine. While this leads to the enrichment of IRF motifs in the promoters of the infection-induced genes in BL/6 macrophages, it does not necessarily mean that genes with the RELB motif are not induced. Indeed, when we examined the expression of the 180 genes induced by IL-4-Fc in BL/6 macrophages that have a NF-κB motif, these genes are also induced by *H. polygyrus* in BL/6 but not BALB/c mice. These results are shown in Fig S1C and Fig.S1D.

This has been added to the results section:

“Lastly, we compared the transcriptional responses to IL-4-Fc and *H. polygyrus* and observed some overlap in the significantly induced genes; however, *H. polygyrus* infection induces a greater number of unique genes, in both BL/6 and BALB/c strains (Fig. S1C). Notably, *H. polygyrus* induces a more distinct transcriptional phenotype in macrophages compared to naïve peritoneal macrophages, likely due to the complex stimuli from the entire parasite response, in contrast to the singular influence of IL-4-Fc. This complexity results in the enrichment of IRF motifs in the promoters of infection-induced genes in BL/6 macrophages, though it does not preclude the induction of genes containing the NF-κB motif. Indeed, our analysis of the 180 genes with NF-κB motifs that are induced by IL-4-

Fc in BL/6 macrophages reveals that these genes are also induced by H. polygyrus, but not in BALB/c mice (Fig. S1D)."

2) I was intrigued by the finding in the scRNA-seq data that in BL/6 TRMs, only a distinct subset of the TRMs appeared to be regulated by NFkB (Figure 6E). As the authors mention, this implies that "differences in strains could potentially be driven by just a small subset of the macrophage population." Is there a way to look specifically at those cells which show the NFkB regulome, and see how these cells differ from the other macrophages transcriptionally (in terms of which genes they express more or less of)? For example, could these cells be highlighted on a UMAP plot, and would they represent a distinct region of the BL/6 macrophage cluster?

As suggested by the Reviewer, we identified the macrophage subset that showed *Rela* and *Relb* regulome activity and plotted these in a separate plot (See below). Notably, they do not form a distinct region in the macrophage cluster. We then performed a differential analysis and compared cells that have either the *Rela* or *Relb* regulome activity compared to all the other macrophages. One gene of particular interest is *Spp1*, which is more highly expressed in a subset of cells with NF- κ B activity. This is now shown in Fig. S6A.

This has been added to the results section:

*"Notably, only a distinct subset of BL/6 TRMs is regulated by the NF- κ B. Hence, differences between strains could potentially be driven by just a small subset of the macrophage population. To investigate this subset further, we identified cells that have either the *Rela* or *Relb* regulon activity and performed a differential analysis with all the other macrophages (Fig. S6A). Notably, *Spp1* is more highly expressed in the subset with NF- κ B regulon activity."*

3) Could the authors speculate in the discussion on the durability of the epigenetic changes induced by IL-4 receptor signaling in the TRMs? For example, how long after the IL-4-Fc treatment is the synergistic response still seen? Are these chromatin accessibility changes induced by IL-4 receptor signaling long lasting, such that exposures to LPS at a much later time point would also show the synergistic response?

To investigate the durability of the epigenetic changes, we performed new experiments and isolated BL/6 peritoneal macrophages 16 days after IL-4-Fc treatment and performed ATAC-seq to determine whether the changes in chromatin accessibility were long-lasting. In contrast to 4801 induced peaks observed 4 days after IL-4-Fc treatment compared to naïve control macrophages, we found no statistically significant differential peaks (FDR > 0.05) at 16 days post-treatment which is also reflected by PCA (Fig. S4E). Regulatory elements containing the NF- κ B motif after IL-4-Fc treatment were no longer accessible following 16 days of IL-4 treatment. An example of IL4+LPS synergistic genes, specifically *Ccl2* and *Ccl7* from Fig. 3E, is shown below.

This has been added to the results section:

*“To investigate the durability of epigenetic changes, we isolated BL/6 peritoneal macrophages 16 days after IL-4-Fc treatment and performed ATAC-seq to determine whether the changes in chromatin accessibility were long-lasting. In contrast to the 4801 induced differential peaks observed 4 days after IL-4-Fc treatment compared to naïve control macrophages, we found no statistically significant differential peaks (FDR \geq 0.05) at 16 days post-treatment and the epigenetic profile is indistinguishable from control macrophages by PCA (Fig. S4E, left). Additionally, regulatory elements containing the NF- κ B motif after IL-4-Fc treatment were no longer accessible following 16 days of IL-4 treatment in specific examples of synergistic genes, such as *Ccl2* and *Ccl7* (Fig. S4E, right). This indicates that the BL/6 synergistic response to LPS stimulation is not long lasting.”*

4) What would the transcriptional and epigenetic responses to IL-4 or IL-4/LPS look like in CB6F1 mice? Would changes in chromatin accessibility after IL-4-Fc happen differently on the BL/6 allele vs. the Balb/c allele for any given gene?

Differences in transcriptional responses in F1 mice between BL/6 and BALB/c alleles is difficult to ascertain because there are very few sequence variants in the transcripts to distinguish between BL/6 and BALB/c, hence we focused on epigenetic responses to IL-4 in F1 mice. The results are shown in Fig. S7 (see below).

Fig S7

This has been added to the results section.

“To determine if the BL/6 and BALB/c alleles are epigenetically regulated differently in the same macrophage of a F1 mouse, we performed a deep ATAC-seq analyses on peritoneal macrophages from IL4-Fc treated F1 mice in comparison to control naïve untreated F1 mice. In order to confidently associate peaks with alleles, we restricted analyses to only reads that mapped perfectly to the genome and also overlap with a sequence variant (between BL/6 and BALB/c genomes), to determine if reads are from BL/6 or BALB/c alleles and to assess differences in regions of open chromatin unique to each parental strain (Fig. S7). Of the 27,431 peaks of naïve macrophages, 1115 are BL/6 specific and 1006 are BALB/c specific. Of the 27,619 peaks of IL-4-Fc treated

macrophages, 1123 are BL/6 specific and 1106 are BALB/c specific (Fig. S7A). Of the 30,927 merged peaks for Control and IL4 conditions, 1270 are induced on the BALB/c allele and 1300 induced on BL/6 allele by IL-4 (Fig. S7B). We did not observe preferential enrichment of the NF- κ B motif in the BL/6-specific IL-4 induced peaks, likely because there are only 623 peaks that do not also change in the BALB/c allele. However, we found that the homeobox motif, such as Meis1, is still more enriched in the BALB/c-specific IL-4 induced peaks, consistent with previous results. Additionally, we observed that the CTCF motif, which organizes genome structure, is more enriched BL/6-specific IL-4 induced peaks (Fig. S7C, D). Hence, the majority of all peaks were cis regulated in both parental alleles and the observed differences between the parental strains could be due to differences in SDTFs protein abundance or activation, rather than sequence variation alone.”

Minor points:

1) Can the authors point to another reference where they have used the IL-4-Fc fusion protein to describe how it was generated in a bit more detail, or describe it here more completely.

This has now been added:

Chenery AL, Rosini S, Parkinson JE, Ajendra J, Herrera JA, Lawless C, Chan BH, Loke P, MacDonald AS, Kadler KE, Sutherland TE, Allen JE. IL-13 deficiency exacerbates lung damage and impairs epithelial-derived type 2 molecules during nematode infection. *Life Sci Alliance*. 2021 Jun 14;4(8):e202001000. doi: 10.26508/lsa.202001000. PMID: 34127548; PMCID: PMC8321663.

This has now been added in the method section.

“A fusion protein of mouse IL-4 with the Fc portion of IgG1 (custom order with Absolute Antibody) was generated to extend half-life with similar effects to IL-4–anti–IL-4 mAb complex. Fc fusion proteins (also known as Fc chimeric fusion proteins, Fc-Igs, Ig-based chimeric fusion proteins and Fc-tag proteins) are composed of the Fc domain of IgG genetically linked to a peptide or protein of interest (mouse IL-4 in this project).”

2) Please add the methodology for the *H. polygyrus* infection experiments. At what day of infection are the mice sacrificed for purification of the peritoneal macrophages? What dose of L3 larvae was given, etc.

This has now been added in the method section.

“For *H. polygyrus* infection experiments, mice were infected with third stage (L3) *H. polygyrus* larvae by oral gavage. 200 Hp larvae was give per mouse. *H. polygyrus* were propagated as previously described. Mice were sacrificed 13 days after *H. polygyrus* infection.”

3) Line 511 of the Discussion has a sentence with what may be an error: “Notably, BMDM experiments also identified EGR2 except NF- κ B as a key...”. Not sure what is meant here, I assume the “EGR2 except” is to be deleted?

We apologize for this error. The sentence now reads as follows:

“Notably, BMDM experiments also identified EGR2 as a key transcription factor influencing the synergistic impact of LPS on IL-4-primed macrophages.”

4) The font in the legends of Fig S3D and Fig S4C should be changed to match the rest of the figures.

This has now been corrected.

Reviewer 2:

(Remarks to the Author)

The manuscript offers a thorough examination of the strain-specific response of tissue-resident memory T cells (TRM) to IL-4 and its synergistic effects with subsequent LPS stimulation. The integration of multiomic approaches, including bulk RNAseq, ATACseq, ChIPseq, and scRNAseq, enriches the analysis and provides valuable insights.

We thank the Reviewer for his positive comments.

To enhance the manuscript's quality further, I recommend addressing the following points:

1. Given BALB/c mice's known propensity for stronger type 2 responses compared to BL/6, it's crucial to investigate whether inherent differences in tonic endogenous IL-4 levels exist between the strains. Such disparities may imprint epigenetic memory on TRM or hematopoietic stem cells (HSC) in the bone marrow, even though bone marrow chimeras are utilized to address cell-intrinsic differences.

The Reviewer makes an excellent point, which is extremely difficult to address. Although we and others have not detected differences in IL-4 or IL-13 levels in naïve BL/6 and BALB/c mice, we cannot rule out inherent differences in tonic signaling between the two strains. The mixed bone marrow chimera experiment was our best effort in trying to resolve cell-intrinsic differences, but this does not rule out tonic signaling in HSCs in the bone marrow prior to the generation of the chimeras. To address this limitation, we have added the following sentence to the discussion:

“It is possible that inherent differences in tonic IL-4 signaling between BALB/c and BL/6 mice may imprint epigenetic memory on the hematopoietic stem cells in the bone marrow, which were transferred into the mixed chimeric mice. One limitation of our studies is that we have not characterized epigenetic differences in the bone marrow between the BALB/c and BL/6 mice.”

2. The assertion that BL/6-specific genes are expressed at higher levels than BALB/c-specific genes, as depicted in FigS1A, might be overstated. Visual assessment of the normalized heatmap expression alone may not suffice to discern these differences accurately (lines 110-111).

As the Reviewer suggested, we have now quantified the read counts of the BL/6 specific and BALB/c specific genes to show that the BL/6-specific genes are expressed at higher levels than BALB/c-specific genes. This is now added to Fig S1B.

Line 110 -112:

“Quantified read counts indicate that BL/6-specific genes are expressed at significantly higher levels compared to BALB/c-specific genes (Fig. S1B).”

3. In FigS2D, the de novo motif analysis should ideally yield multiple motifs instead of just one. Providing the entire motif list and justifying the selection of specific motifs for demonstration would enhance clarity and transparency.

As suggested by the Reviewer, we have now provided the top20 motif list for Fig.S2D.

C57BL/6	P-value	Best Match	BALB/c	P-value	Best Match
	1e-12	IRF2		1e-13	MEIS1
	1e-10	Rbpj1		1e-10	PB019.1
	1e-10	ZNF682		1e-10	IRF1
	1e-9	ZKSCAN5		1e-10	IKZF1
	1e-9	ZFP57		1e-10	PB0098.1
	1e-9	Zic(Zf)		1e-9	RUNX2
	1e-9	PB0108		1e-9	ZNF768
	1e-9	E2F6		1e-9	PB0198.1
	1e-9	IKZF1		1e-9	PU.1
	1e-8	PB0199.1		1e-9	NFE2
	1e-8	Erra		1e-9	PH0044.1
	1e-8	NFIC		1e-8	NFIC
	1e-7	PB0180.1		1e-8	HOXD3
	1e-7	EWS:ERG		1e-7	NFIX
	1e-6	ZNF341		1e-7	MZF1
	1e-6	PB0071.1		1e-7	STAT3
	1e-6	GFY		1e-6	SOX18
	1e-5	PB0156		1e-6	PB0203.1
	1e-5	ZBTB32		1e-6	FOXN3
	1e-4	REL		1e-5	MEIS1

4. Besides examining chemokine genes in Fig3E, the authors should also assess proinflammatory genes such as IL-1, TNF, and IL6, given IL-4's reported synergistic upregulation of immune response, as indicated in the C2 GO analysis (FigS3E).

As suggested by the Reviewer, we also assessed the expression of the proinflammatory genes IL-1a, IL-1b, TNF and IL-6 in this experiment and we do not observe synergistic induction of these genes by IL-4+LPS indicating that only specific inflammatory molecules are regulated synergistically in peritoneal macrophages. This is now included in Fig. S3F.

Line 265-268:

“Other proinflammatory genes (Il1a, Il1b, Il6, and Tnf) show no synergistic effect in either mouse strain, indicating only a subset of genes exhibit a synergistic response like those in the C2 cluster (Fig. S3F).”

Similarly, including the full motif list as supplementary data for the de novo analysis in Fig3G would provide a more comprehensive understanding.

The full motif list (Fig. S3G) has been included as supplementary data for Fig. 3G.

C57BL/6	P-value Best Match	BALB/c	P-value Best Match
	1e-610 Sfp1		1e-418 Sfp1
	1e-177 IRF3		1e-112 KLF1
	1e-127 CEBPA		1e-96 BATF
	1e-118 KLF1		1e-81 NFIL3
	1e-117 JunB		1e-54 MITF
	1e-64 PRDM4		1e-48 Dux
	1e-51 FOSL1:JUND		1e-48 IRF3
	1e-43 TFEB		1e-26 RUNX
	1e-37 NFY(CCAAT)		1e-23 Mafb
	1e-29 RUNX-AML		1e-23 ZNF449
	1e-26 EKLF		1e-22 MEF2D
	1e-24 NFkB-p65		1e-20 GFY
	1e-23 BORIS		1e-19 CEBPD
	1e-19 MEF2A		1e-18 BACH1
	1e-16 NFIL3		1e-17 NFYA
	1e-12 Bcl6b		1e-16 CREB1
	1e-10 Ap4		1e-14 BORIS
	1e-10 ELF4		1e-12 PB0059
	1e-9 GFX		1e-10 NRF
	1e-9 PB0117.1		1e-8 POL010.1

5.The adapted murine Sepsis score (MSS) requires definition, and detailed scoring criteria should be included in the materials and methods section to facilitate assessment of its validity.

Definition of MSS score based on the paper (Mai, S.H.C. et al. Body temperature and mouse scoring systems as surrogate markers of death in cecal ligation and puncture sepsis. *Intens Care Med Exp* 6 (2018).) was added in the method.

6.In Fig7D, presenting actual cell counts in addition to percentages would offer more informative insights, especially considering the potential surge in neutrophil counts following LPS treatment. It's also essential to examine basal cell counts in the IL-4 treated group without LPS treatment to ascertain the impact of IL-4 on peritoneal macrophage expansion.

We now include cell count data in Fig. S6E. However, while we loaded the same number of cells for the 10X analysis, there is variation in the overall recovery rates. There is certainly an expansion of peritoneal macrophages by IL-4 and there is also better recovery compared to LPS treated peritoneal macrophages.

This has now been added in the result section.

“IL-4 pre-treatment expanded peritoneal macrophages and reduced the loss of macrophages after LPS treatment (Fig. 7G and Fig. S6E).”

7. The interpretation of macrophage heterogeneity and clustering, along with associated regulons, requires clarification. For instance, while both C2 and C4 appear depleted after LPS exposure, only C2 is discussed. Similarly, although C6 and C9 seem to increase in the IL-4+LPS group, only C6 is addressed.

We found that C8 instead of C9 is increased in the IL-4+LPS group. We have provided more discussion of macrophage heterogeneity and clustering in the results section.

This discuss has now been added in the result section

“Some macrophage clusters experience significant depletion after LPS exposure, such as Cluster 2 and Cluster 4. By SCENIC analysis, the mostly depleted clusters are enriched with TRM-specific lineage and/or functional markers. For example, Cluster 2 and Cluster 4 macrophages are distinguished by the activity of Gata6, Klf, and Fli1 regulons (Fig. 7H, Fig. S6D). In Cluster 4, some macrophage activation-related transcription factors in TRMs are enriched, such as Foxp1 and Atf7. Cluster 6 macrophages are more abundant in the IL-4+LPS treatment group compared to the LPS-only group, particularly from the BL/6 background (Fig. S6D). SCENIC analysis of Cluster 6 reveals an enrichment of the NF-κB (Rel) regulon. Additionally, we found that clusters 3, 5, and 8 are also increased in the LPS + IL-4 group, especially in BL/6 mice. Among the motif analysis after SCENIC, we found that the Ezh2 regulon is more enriched in Cluster 8. Ezh2 has been shown to physically interact with RelA via the transactivation domain and can co-activate the transcription of a subset of NF-κB target genes with RelA. Furthermore, we found that the Nfkb1 and Stats regulons are enriched in clusters 3 and 5, which are also more induced in the LPS + IL-4 group in BL/6 mice. All clusters (3, 5, 6 and 8) increase in response to LPS + IL-4 treatment in BL/6, indicating NF-κB regulation. This is consistent with our conclusion that NF-κB is involved in a synergistic response in BL/6 mice.”

8. Regarding the discussion on strain-specific effects in regulons (Fig7H and Fig7I), it appears that some information may be lost, or combined data rather than strain-specific

data may have been utilized for the analysis. Providing details about the analysis methodology would help clarify this aspect.

We have clarified the discussion on strain specific regulons in the analysis methods that combined data was used.

Line 471-473:

“This analysis integrated data from both BL/6 and BALB/c mice, allowing for a comprehensive comparison of the clustering and treatment conditions.”

Minor comments:

1. There is a punctuation error in line 44.

This has been corrected. Thank you for spotting the error.

2. Recent reports on IL-4-induced trained immunity (PMID: 37291433, 37149865, 36173104) are relevant and should be included for discussion.

A new paragraph has been added to the discussion to include these reports:

“IL-4 can remodel the BL/6 epigenome to expose NF-κB binding sites, contributing to a distinct and robust inflammatory response to LPS, akin to trained immunity. Trained immunity, where innate immune cells exhibit heightened responses to subsequent stimuli is important for how the immune system can adapt and enhance its responses over time. How genetic background shapes immune responses may depend on how exposure to IL-4 could remodel epigenetic pathways in macrophages or monocytes, to increase capacity to respond to future challenges.”

Reviewer #3

(Remarks to the Author):

Zhao and colleagues present a significant collection of data indicating strain-specific difference in tissue resident macrophages in response to IL-4 mediated activation. There data data clearly show in vivo IL-4-Fc causes significant strain specific transcriptomic changes in BL/6 and Balb/C mice, with a greater effect in BL/6 mice. H polygyrus confirms an alternative macrophage activation result that is greater in BL/6 mice cf. Balb/c. IL-4 increased inducer in BL/6 greater than Balb/c. In Balb/c IL-4 barely affects LPS signaling, yet it change BL/6 responses. The work is of significance to the field and supports varying immune "prone" responses particularly in Balb/c and C57Bl/6 mice. However, there remains some importance questions that if answered could potentially strengthen this manuscript.

Major comments:

1. Were the upregulated genes in response to IL-4 stimulation in TRMs and BMDMs similar? This was not discussed and may provide clarity on conserved responses across species and cellular compartments/subtypes.

We have now compared the IL-4 upregulated genes in TRMs and BMDMs. These results show that there is considerable variation in responses between both strains and macrophage subtypes. These data is now included in Fig S1E.

In the results section we add:

“Additionally, when we compared IL-4 upregulated genes in TRMs and BMDMs, there is considerable variation in responses between both strains and macrophage subtypes indicating differences in IL-4 responsiveness by compartment as well as genetic background (Fig. S1E).”

2. The authors clearly demonstrated that IL-4-upregulated genes are greater in BL/6/. However, GO terms are not very informative on function (Figure S1B). While greater transcriptional function is reported, what is the function? This should be discussed in better detail to extrapolate function.

We agree that the GO terms are not very informative on function. Indeed, we decided that this analysis was not particularly useful for the overall message of the manuscript and could be misleading. Hence, we have decided to remove this Supplemental Figure from the manuscript.

3. The biggest weakness is that the sole focus on macrophage epigenome, yet the treatment with IL-4 or LPS is in vivo. How are the changes in macrophage function not due to the influence of other cells namely T cells/B cells responding to IL-4, that then drive the macrophage response?
4. Is the responsiveness to IL-4 augmented in macrophages from BL/6 mice because they are more Th1 prone and likely receiving basal/tonic Th1 cytokine stimulation as opposed to balb/c mice that may be desensitised to IL-4 stimulation?

The Reviewer makes an excellent point, which was also raised by Reviewer 2 and is extremely difficult to address. Although we have performed mixed bone marrow chimera experiments into F1 mice with BL/6 and BALB/c macrophages that are in the same mouse tissue environment, so in theory they should be responding intrinsically to IL-4 in the context of the same cells, we cannot rule out that the transfer of B cells from BL/6 and BALB/c mice could also interact differently with the macrophages in the same tissue environment. T cells were depleted from the bone marrow cells before transfer.

Additionally, as Reviewer 2 also noted, although we and others have not detected differences in IL-4 or IL-13 levels in naïve BL/6 and BALB/c mice, we cannot rule out

inherent differences in tonic signaling between the two strains. The mixed bone marrow chimera experiment was our best effort in trying to resolve cell-intrinsic differences, but this does not rule out tonic signaling in HSCs in the bone marrow prior to the generation of the chimeras. To address this limitation, we have added the following sentence to the discussion:

“It is possible that inherent differences in tonic IL-4 signaling between BALB/c and BL/6 mice may imprint epigenetic memory on the hematopoietic stem cells in the bone marrow, which were transferred into the mixed chimeric mice. One limitation of our studies is that we have not characterized epigenetic differences in the bone marrow between the BALB/c and BL/6 mice. Additionally, bone marrow transfers would also bring B cells from BL/6 and BALB/c mice into the same F1 environment, and we cannot rule out that this may affect macrophage intrinsic responses.”

The transcriptome while important and predictive requires validation. Is the function of the macrophages from each species different *in vitro* in response to IL-4 +/- LPS?

In response to the Reviewer’s question, we performed additional experiments to show that BALB/c and BL/6 peritoneal macrophages could behave different functionally in response to IL-4 +/- LPS priming *in vitro*, but these results were not conclusive. For this experiment, we employed *Toxoplasma gondii* as an intracellular parasite for which replication in macrophages is inhibited by nitric oxide.
(<https://www.sciencedirect.com/science/article/pii/S0171298599800643>).

Peritoneal F4/80+ macrophages from IL-4-Fc treated mice (pooled from 5 animals/group) were incubated with or without LPS (100 ng/ml) for 4hrs in 96-flat-bottom plates before infection with the RH strain of *T. gondii* (MOI 1:1). After an additional 24hr incubation, we examined the frequency of infected cells in cultures treated with or without LPS.

In the absence of LPS, 100% of macrophages from either strain of mice were infected harboring >4 parasites/cell. Despite this high infection rate, LPS-treated cultures contained a significant proportion of uninfected cells which was slightly higher in C57BL/6 (32%) when compared to BALB/c (27%) macrophages. Moreover, in the C57BL/6 cultures more cells 32% vs. 28% had less than 4 parasites/cell.

While this preliminary experiment indicates an advantage of IL-4 +/- LPS primed C57/BL/6 macrophages after *in vitro* infection with *T. gondii*. We hope the reviewer agrees that significance of this difference *in vitro* between the macrophages from C57BL/6 and BALB/c after IL-4 +/- LPS priming is quite subtle, and we are not confident of including this result in the manuscript.

It is possible that further optimization at various MOI, and different concentrations of LPS and additional time points would yield a clearer and more significant result, but this would we believe be beyond the scope of the manuscript and significantly delay publication.

In the discussion, we have added the following sentences:

“Preliminary experiments indicate that BL/6 macrophages may have an advantage in an in vitro setting against infection with T. gondii (data not shown), but further experiments are required to draw firmer conclusions.”

Reviewer #4

(Remarks to the Author):

The authors showed through a comprehensive set of bulk RNA and single cell RNA experiments that TRMs from BL/6 mice have a more pronounced response to IL-4 than TRMs from BALB/c mice. Through computational predictive analyses, 1-dimensional and 3-dimensional chromatin analyses the authors showed the more pronounced response in BL/6 mice compared to BALB/c mice arose from epigenomic remodelling mediated by NF-kB binding sites that were intrinsic to the cells, irrespective of the tissue environment the cells were found in. Lastly, the authors showed the synergistic response of IL-4 and LPS treatment could also be attributed to epigenomic remodelling mediated by NF-kB.

We thank the Reviewer for their appreciation of the main points of this study.

Comments:

1. Lines 111-112: “Gene ontology (GO) analysis of BL/6 or BALB/c specific genes indicates some genotype specific functional differences in chemotaxis and cell migration (Fig S1B).” – However, the GO terms in figure S1B for both strains are related to cell locomotion, taxis and migration. Since GO terms are hierarchical and can belong to the same “family” of terms despite sometimes different wording, could the author please explain/elaborate on these genotype specific differences?

As noted above in our response to Reviewer 3, we decided that the GO analysis is not very informative on function. Indeed, we decided that this analysis was not particularly useful for the overall message of the manuscript and could be misleading. Hence, we have decided to remove this Supplemental Figure from the manuscript.

2. Throughout the text, the authors focus on upregulated genes, despite evidence from their figures also showing a number of downregulated genes, including the identification of motifs for those genes. Can the authors please explain why they have chosen to focus only on upregulated genes?

We have now added this sentence to the discussion:

“In this manuscript we have focused on upregulated genes because the mechanisms of transcriptional activators (vs repressors) in relation to epigenetic changes is better understood. Future studies will investigate genes that are specifically downregulated by IL-4 in BL/6 and BALB/c macrophages.”

3. In the text and figures, the authors said they infected BL/6 and BALB/C mice with Heligomosomoides polygyrus (H. polygyrus), however, the methodology used for these experiments have not been included in the methods section of the manuscript.

We apologize for this oversight and have now included these experiments in the methods.

4. Lines 156-159: “Notably, BALB/c macrophages have more strain specific constitutive regions (n = 18,336 in BALB/c versus n = 3,731 in BL/6), whereas BL/6 macrophages have more IL-4 inducible accessible regions (n = 4,801 in BL/6 versus n = 3,660 in BALB/c) (Fig. 2A and 2B).” The difference in strain specific constitutively active regions is quite pronounced between the strains, whereas the differences following IL-4 stimulation are much less so. Are these differences of ~1200 regions significant?

The Reviewer makes an excellent point and this difference is probably not significant. Hence, we have now clarified and changed the language to:

“whereas there is only a small difference in IL-4 inducible accessible regions (n = 4,801 in BL/6 versus n = 3,660 in BALB/c) (Fig. 2A and 2B).”

5. In figure S3B, there are no labels for which GO terms are from BALBc and which are from BL/6 mice.

We apologize for this oversight and labels are now added.

6. Lines 381-382: “Notably, only a distinct subset of BL/6 TRMs is regulated by the NF-κB. Hence, differences between strains could potentially be driven by just a small subset of the macrophage population.”. It isn’t until the end of the following results section (lines 444-457) that the authors expand on this point leaving the reader unsure of whether they will find out more about this specific subset.

We have now expanded upon this analysis and provided more information on this subset. We identified the macrophage subset that showed Rela and Relb regulome activity and plotted these in a separate plot (See below). We then performed a differential analysis and compared cells that have either the Rela or Relb regulome activity compared to all the other macrophages. One gene of particular interest is Spp1, which is more highly expressed in a subset of cells with NF-κB activity.

This results section has been edited to the following:

“Notably, only a distinct subset of BL/6 TRMs is regulated by the NF-κB. Hence, differences between strains could potentially be driven by just a small subset of the macrophage population. To investigate this subset further, we identified cells that have either the Relα or Relβ regulon activity and performed a differential analysis with all the other macrophages (Fig. S6A). Notably, Spp1 is more highly expressed in the subset with NF-κB regulon activity.”

7. Lines 411-412: “We used an adapted murine sepsis score (MSS) to evaluate the effects of sub-lethal dose LPS challenge (Fig. 7A).” Please provide more information in the methods on how the MSS score was adapted for future reproducibility.

The reference and more detailed methods have been added.

8. For the majority of the figures, the authors have presented BL/6 data before BALB/c data, however there are some inconsistencies such as in Fig 7.A and 7.E where BALB/c data is presented before BL/6 data. Please fix.

This has been fixed.

9. Lines 428-429: “To identify differential peaks among groups, our focus was exclusively on highly induced genes (logFC threshold ≥ 0.25)”. Did the authors possibly mean a log2 fold change greater than or equal to 0.25 which would translate to a fold change of 2 or higher?

We thank the Reviewer for spotting this error as we were referring to gene expression and not accessible peaks. Also, we were not focusing on highly induced genes. This sentence has been rewritten to:

“We identified genes that are differentially regulated between groups (logFC threshold ≥ 0.25).”

10. Discussion feels scattered with paragraphs jumping a lot and in certain instances displaying conversational language. There are also a lot of acronyms, with no list of abbreviations which reduce the readability of the text. Additionally, in several places, sentence structures are awkward and could be revisited to improve readability.

As suggested by the Reviewer, we have reorganized and rewritten the discussion significantly to improve the readability.

11. Authors need to justify the use of only male mice in their work.

The usage of male mice in these experiments is for historical reasons in that previous scRNA-seq experiments and other genomics experiments that the lab has performed started in male mice, which were more prone to metabolic diseases on the BL/6 background. We wanted to retain the possibility of meta-analysis of the scRNA-seq data

with previous studies. As required by the journal, we have now specified that we used male mice in the abstract.

We have added this to the discussion.

“An additional caveat of these studies is that we do not distinguish between embryonically and bone marrow derived resident peritoneal macrophages and have only data on male mice, while there are gender specific differences at the rate by which monocyte-derived F4/80hi macrophages displace the embryonic population with age (Bain, C., Hawley, C., Garner, H. et al. Long-lived self-renewing bone marrow-derived macrophages displace embryo-derived cells to inhabit adult serous cavities. Nat Commun 7, ncomms11852 (2016). <https://doi.org/10.1038/ncomms11852>.)”

Point-by-point response to the reviewers' comments

REVIEWERS' COMMENTS

Reviewer #1 (Remarks to the Author):

In this revision the authors have significantly improved the manuscript. In particular I appreciate the new experiment testing the durability of the response to IL-4-Fc and the experiment in the CB6F1 mice. I also appreciate the comments of the other Reviewers, which have also improved the manuscript. I have only two minor comments based on two of the new supplementary figures.

1) In new Fig S1B (in response to a critique from Reviewer 2), the text now reads "Quantified read counts indicate that BL/6-specific genes are expressed at significantly higher levels compared to BALB/c-specific genes (Fig. S1B). Could the authors add the information on the statistical test used to make this claim, and show p values in some way on the figure (perhaps as asterisks) for the relevant comparisons made.

We thank the Reviewer for this suggestion and have now include an indication of p-values as asterisks in Fig. S1B.

2) In new Fig S1D, in response to my critique, I think the x axis labels are missing some detail. Are the different columns supposed to represent Ctrl and IL-4 or Ctrl and H. polygyrus, in both BL/6 and Balb/c?

We thank the Reviewer for spotting our labeling error. We have now corrected the labels on Fig. S1D.

Reviewer #2 (Remarks to the Author):

I acknowledge the efforts of the authors to improve the manuscript and their responses to my previous questions. I have no further questions, and the manuscript appears to be in good shape.

Reviewer #3 (Remarks to the Author):

The authors have addressed or attempted to address my concerns with relevant additional data and experiments that may have been inconclusive.

I have no further comments.

Reviewer #4 (Remarks to the Author):

Thank you to the authors for answering the reviewer questions thoughtfully and

thoroughly. The manuscript is much improved and the findings quite interesting.